# Universal Approximation For Log-concave Distributions Using Well-conditioned Normalizing Flows

**Holden Lee**
Mathematics Department
Duke University
Durham, NC 27708
holden.lee@duke.edu

**Chirag Pabbaraju**
Computer Science Department
Stanford University
Stanford, CA 94305
cpabbara@cs.stanford.edu

**Anish Sevekari**
Department of Mathematical Sciences
Carnegie Mellon University
Pittsburgh, PA 15213
asevekar@andrew.cmu.edu

**Andrej Risteski**
Machine Learning Department
Carnegie Mellon University
Pittsburgh, PA 15213
aristesk@andrew.cmu.edu

## Abstract

Normalizing flows are a widely used class of latent-variable generative models with a tractable likelihood. Affine-coupling models [Dinh et al., 2014, 2016] are a particularly common type of normalizing flows, for which the Jacobian of the latent-to-observable-variable transformation is triangular, allowing the likelihood to be computed in linear time. Despite the widespread usage of affine couplings, the special structure of the architecture makes understanding their representational power challenging. The question of universal approximation was only recently resolved by three parallel papers [Huang et al., 2020, Zhang et al., 2020, Koehler et al., 2020] – who showed reasonably regular distributions can be approximated arbitrarily well using affine couplings—albeit with networks with a nearly-singular Jacobian. As ill-conditioned Jacobians are an obstacle for likelihood-based training, the fundamental question remains: which distributions can be approximated using *well-conditioned* affine coupling flows?

In this paper, we show that any *log-concave* distribution can be approximated using well-conditioned affine-coupling flows. In terms of proof techniques, we uncover and leverage deep connections between affine coupling architectures, underdamped Langevin dynamics (a stochastic differential equation often used to sample from Gibbs measures) and Hénon maps (a structured dynamical system that appears in the study of symplectic diffeomorphisms). Our results also inform the practice of training affine couplings: we approximate a padded version of the input distribution with iid Gaussians—a strategy which Koehler et al. [2020] empirically observed to result in better-conditioned flows, but had hitherto no theoretical grounding. Our proof can thus be seen as providing theoretical evidence for the benefits of Gaussian padding when training normalizing flows.

## 1 Introduction

Normalizing flows [Dinh et al., 2014, Rezende and Mohamed, 2015] are a class of generative models parametrizing a distribution in $\mathbb{R}^d$ as the pushfoward of a simple distribution (e.g. Gaussian) through an invertible map $g_\theta : \mathbb{R}^d \to \mathbb{R}^d$ with trainable parameter $\theta$. The fact that $g_\theta$ is invertible allows us to

35th Conference on Neural Information Processing Systems (NeurIPS 2021).

write down an explicit expression for the density of a point $x$ through the change-of-variables formula, namely $p_\theta(x) = \phi(g_\theta^{-1}(x))\det(Dg_\theta^{-1}(x))$, where $\phi$ denotes the density of the standard Gaussian. For different choices of parametric families for $g_\theta$, one gets different families of normalizing flows, e.g. affine coupling flows [Dinh et al., 2014, 2016, Kingma and Dhariwal, 2018], Gaussianization flows [Meng et al., 2020], sum-of-squares polynomial flows [Jaini et al., 2019].

In this paper we focus on affine coupling flows – arguably the family that has been most successfully scaled up to high resolution datasets [Kingma and Dhariwal, 2018]. The parametrization of $g_\theta$ is chosen to be a composition of so-called *affine coupling blocks*, which are maps $f : \mathbb{R}^d \to \mathbb{R}^d$, s.t. $f(x_S, x_{[d]\setminus S}) = (x_S, x_{[d]\setminus S} \odot s(x_S) + t(x_S))$, where $\odot$ denotes entrywise multiplication and $s, t$ are (typically simple) neural networks. The choice of parametrization is motivated by the fact that the Jacobian of each affine block is triangular, so that the determinant can be calculated in linear time.

Despite the empirical success of this architecture, theoretical understanding remains elusive. The most basic questions revolve around the representational power of such models. Even the question of universal approximation was only recently answered by three concurrent papers [Huang et al., 2020, Zhang et al., 2020, Koehler et al., 2020]—though in a less-than-satisfactory manner, in light of how normalizing flows are trained. Namely, Huang et al. [2020], Zhang et al. [2020] show that any (reasonably well-behaved) distribution $p$, once padded with zeros and treated as a distribution in $\mathbb{R}^{d+d'}$, can be arbitrarily closely approximated by an affine coupling flow. While such padding can be operationalized as an algorithm by padding the training image with zeros, it is never done in practice, as it results in an ill-conditioned Jacobian. This is expected, as the map that always sends the last $d'$ coordinates to 0 is not injective. Koehler et al. [2020] prove universal approximation without padding; however their construction *also* gives rise to a poorly conditioned Jacobian: namely, to approximate a distribution $p$ to within accuracy $\epsilon$ in the Wasserstein-1 distance, the Jacobian of the network they construct will have smallest singular value on the order of $\epsilon$.

Importantly, for all these constructions, the condition number of the resulting affine coupling map is poor *no matter how nice the underlying distribution it's trying to approximate is*. In other words, the source of this phenomenon isn't that the underlying distribution is low-dimensional or otherwise degenerate. Thus the question arises:

**Question:** *Can well-behaved distributions be approximated by an affine coupling flow with a well-conditioned Jacobian?*

In this paper, we answer the above question in the affirmative for a broad class of distributions – log-concave distributions – if we pad the input distribution not with zeroes, but with independent Gaussians. This gives theoretical grounding of an empirical observation in Koehler et al. [2020] that Gaussian padding works better than zero-padding, as well as no padding.

The practical relevance of this question is in providing guidance on the type of distributions we can hope to fit via training using an affine coupling flow. Theoretically, our techniques uncover some deep connections between affine coupling flows and two other (seeming unrelated) areas of mathematics: *stochastic differential equations* (more precisely *underdamped Langevin dynamics*, a "momentum" variant of the standard overdamped Langevin dynamics) and *dynamical systems* (more precisely, a family of dynamical systems called *Hénon-like maps*).

## 2 Overview of results

In order to state our main result, we introduce some notation and definitions.

### 2.1 Notation

**Definition 1.** An *affine coupling block* is a map $f : \mathbb{R}^d \to \mathbb{R}^d$, s.t. $f(x_S, x_{[d]\setminus S}) = (x_S, x_{[d]\setminus S} \odot s(x_S) + t(x_S))$ for some set of coordinates $S$, where $\odot$ denotes entrywise multiplication and $s, t$ are trainable (generally non-linear) functions. An *affine coupling network* is a finite sequence of affine coupling blocks. Note that the partition $(S, [d] \setminus S)$, as well as $s, t$ may be different between blocks. We say that the non-linearities are in a class $\mathcal{F}$ (e.g., neural networks, polynomials, etc.) if $s, t \in \mathcal{F}$.

The appeal of affine coupling networks comes from the fact that the Jacobian of each affine block is triangular, so calculating the determinant is a linear-time operation.

We will be interested in the *conditioning* of $f$—that is, an upper bound on the largest singular value $\sigma_{\max}(Df)$ and lower bound on the smallest singular value $\sigma_{\min}(Df)$ of the Jacobian $Df$ of $f$. Note that this is a slight abuse of nomenclature – most of the time, "condition number" refers to the ratio of the largest and smallest singular value. As training a normalizing flow involves evaluating $\det(Df)$, we in fact want to ensure that neither the smallest nor largest singular values are extreme.

The class of distributions we will focus on approximating via affine coupling flows is *log-concave* distributions:

**Definition 2.** A distribution $p : \mathbb{R}^d \rightarrow \mathbb{R}^+, p(x) \propto e^{-U(x)}$ is *log-concave* if $\nabla^2 U(x) = -\nabla^2 \ln p(x) \succeq 0$.

Log-concave distributions are typically used to model distributions with Gaussian-like tail behavior. What we will leverage about this class of distributions is that a special stochastic differential equation (SDE), called *underdamped Langevin dynamics*, is well-behaved in an analytic sense. Finally, we recall the definitions of positive definite matrices and Wasserstein distance, and introduce a notation for truncated distributions.

**Definition 3.** We say that a symmetric matrix is *positive semidefinite (PSD)* if all of its eigenvalues are non-negative. For symmetric matrices $A, B$, we write $A \succeq B$ if and only if $A - B$ is PSD.

**Definition 4.** Given two probability measures $\mu, \nu$ over a metric space $(M, d)$, the *Wasserstein*-1 distance between them, denoted $W_1(\mu, \nu)$, is defined as

$$W_1(\mu, \nu) = \inf_{\gamma \in \Gamma(\mu, \nu)} \int_{M \times M} d(x, y) \, d\gamma(x, y)$$

where $\Gamma(\mu, \nu)$ is the set of couplings, i.e. measures on $M \times M$ with marginals $\mu, \nu$ respectively. For two probability *distributions* $p, q$, we denote by $W_1(p, q)$ the Wasserstein-1 distance between their associated measures. In this paper, we set $M = \mathbb{R}^d$ and $d(x, y) = \|x - y\|_2$.

**Definition 5.** Given a distribution $q$ and a compact set $\mathcal{C}$, we denote by $q|_{\mathcal{C}}$ the distribution $q$ truncated to the set $\mathcal{C}$. The truncated measure is defined as $q|_{\mathcal{C}}(A) = \frac{1}{q(\mathcal{C})} q(A \cap \mathcal{C})$.

## 2.2 Main result

Our main result states that we can approximate any log-concave distribution in Wasserstein-1 distance by a *well-conditioned* affine-coupling flow network. Precisely, we show:

**Theorem 1.** *Let $p(x) : \mathbb{R}^d \rightarrow \mathbb{R}^+$ be of the form $p(x) \propto e^{-U(x)}$, such that:*

  1. *$U \in C^2$, i.e., $\nabla^2 U(x)$ exists and is continuous.*
  2. *$\ln p$ satisfies $\mathrm{I}_d \preceq -\nabla^2 \ln p(x) \preceq \kappa \mathrm{I}_d$.*

*Furthermore, let $p_0 := p \times \mathcal{N}(0, \mathrm{I}_d)$. Then, for every $\epsilon > 0$, there exists a compact set $\mathcal{C} \subset \mathbb{R}^{2d}$ and an invertible affine-coupling network $f : \mathbb{R}^{2d} \rightarrow \mathbb{R}^{2d}$ with polynomial non-linearities, such that*

$$W_1(f_\#(\mathcal{N}(0, \mathrm{I}_{2d})|_{\mathcal{C}}), p_0) \leq \epsilon.$$

*Furthermore, the map defined by this affine-coupling network $f$ is well conditioned over $\mathcal{C}$, that is, there are positive constants $A(\kappa), B(\kappa) = \kappa^{O(1)}$ such that for any unit vector $w$,*

$$A(\kappa) \leq \|D_w f(x, v)\| \leq B(\kappa)$$

*for all $(x, v) \in \mathcal{C}$, where $D_w$ is the directional derivative in the direction $w$. In particular, the condition number of $Df(x, v)$ is bounded by $\frac{B(\kappa)}{A(\kappa)} = \kappa^{O(1)}$ for all $(x, v) \in \mathcal{C}$.*

We make several remarks regarding the statement of the theorem:

*Remark* 1. The Gaussian padding (i.e. setting $p_0 = p \times \mathcal{N}(0, \mathrm{I}_d)$) is essential for our proofs. All the other prior works on the universal approximation properties of normalizing flows (with or without padding) result in ill-conditioned affine coupling networks. This gives theoretical backing of empirical observations on the benefits of Gaussian padding in Koehler et al. [2020].

*Remark* 2. The choice of non-linearities $s, t$ being polynomials is for the sake of convenience in our proofs. Using standard universal approximation results, they can also be chosen to be neural networks with a smooth activation function.

*Remark* 3. The Jacobian $Df$ has both upper-bounded largest singular value, and lower-bounded smallest singular value—which of course bounds the determinant $\det(Df)$. As remarked in Section 2.1, merely bounding the ratio of the two quantities would not suffice for this. Moreover, the bound we prove *only* depends on properties of the distribution (i.e., $\kappa$), and does not worsen as $\epsilon \to 0$, in contrast to Koehler et al. [2020].

*Remark* 4. The region $\mathcal{C}$ where the pushforward of the Gaussian through $f$ and $p_0$ are close is introduced solely for technical reasons—essentially, standard results in analysis for approximating smooth functions by polynomials can only be used if the approximation needs to hold on a compact set. Note that $\mathcal{C}$ can be made arbitrarily large by making $\epsilon$ arbitrarily small.

*Remark* 5. We do not provide an explicit computation of the number of affine coupling blocks in the constructed network, although a bound of $\mathrm{polylog}(\epsilon)/\epsilon^{O(k)}$ can be extracted from our proofs.

*Remark* 6. Our proof also implies a well-conditioned universal approximation result for other related normalizing flow models. Lemma 1 proves that the flow map of underdamped Langevin dynamics is well conditioned for all $t \in [0, T]$. However, as indicated in Chen et al. [2018], underdamped Langevin dynamics is a continuous normalizing flow, thus the claim applies to such flows as well. Similarly, the particular affine coupling layers we construct in eq. (13) also form a residual block, so the claim also holds for residual flows [Behrmann et al., 2018].

# 3    Related Work

The landscape of normalizing flow models is rather rich. The inception of the ideas was in Rezende and Mohamed [2015] and Dinh et al. [2014], and in recent years, an immense amount of research has been dedicated to developing different architectures of normalizing flows. The focus of this paper are affine coupling flows, which were introduced in Dinh et al. [2014], introduced the idea of using pushforward maps with triangular Jacobians for computational efficiency. This was further developed in Dinh et al. [2016] and culminated in Kingma and Dhariwal [2018], who introduced 1x1 convolutions in the affine coupling framework to allow for "trainable" choices of partitions. We note, there have been variants of normalizing flows in which the Jacobian is non-triangular, e.g. [Grathwohl et al., 2018, Dupont et al., 2019, Behrmann et al., 2018], but these models still don't scale beyond datasets the size of CIFAR-10.

In terms of theoretical results, the most closely related works are Huang et al. [2020], Zhang et al. [2020], Koehler et al. [2020]. The former two show universal approximation of affine couplings—albeit if the input is padded with zeros. This of course results in maps with singular Jacobians, which is why this strategy isn't used in practice. Koehler et al. [2020] show universal approximation without padding—though their constructions results in a flow model with condition number $1/\epsilon$ to get approximation $\epsilon$ in the Wasserstein sense, regardless of how well-behaved the distribution to be approximated is. Furthemore, Koehler et al. [2020] provide some empirical evidence that padding with iid Gaussians (as in our paper) is better than both zero padding (as in Huang et al. [2020], Zhang et al. [2020]) and no padding on small-scale data.

# 4    Preliminaries

Our techniques leverage tools from stochastic differential equations and dynamical systems. We briefly survey the relevant results.

## 4.1    Langevin Dynamics

Broadly, Langevin diffusions are families of stochastic differential equations (SDEs) which are most frequently used as algorithmic tools for sampling from distributions specified up to a constant of proportionality. They have also recently received a lot of attention as tools for designing generative models [Song and Ermon, 2019, Song et al., 2020].

In this paper, we will only make use of *underdamped Langevin dynamics*, a momentum-like analogue of the more familiar *overdamped Langevin dynamics*, defined below. Our construction will involve simulating underdamped Langevin dynamics using affine coupling blocks.

**Definition 6** (Underdamped Langevin Dynamics). *Underdamped Langevin dynamics* with potential $U$ and parameters $\zeta, \gamma$ is the pair of SDEs

$$\begin{cases} dx_t &= -\zeta v_t dt \\ dv_t &= -\gamma \zeta v_t dt - \nabla U(x_t) dt + \sqrt{2\gamma}\, dB_t. \end{cases} \tag{1}$$

The stationary distribution of the SDEs (limiting distribution as $t \to \infty$) is given by $p^*(x,v) \propto e^{-U(x) - \frac{\zeta}{2}\|v\|^2}$.

The variable $v_t$ can be viewed as a "velocity" variable and $x_t$ as a "position" variable – in that sense, the above SDE is an analogue to momentum methods in optimization.

The convergence of (1) can be bounded when the distribution $p(x) \propto \exp(-U(x))$ satisfies an analytic condition, namely has a bounded *log-Sobolev* constant. Though we don't use the log-Sobolev constant in any substantive manner in this paper, we include the definition for completeness.

**Definition 7.** A distribution $p : \mathbb{R}^d \to \mathbb{R}^+$ satisfies a *log-Sobolev* inequality with constant $C > 0$ if $\forall g : \mathbb{R}^d \to \mathbb{R}$, s.t. $g^2, g^2|\log g^2| \in L^1(p)$, we have

$$\mathbb{E}_p[g^2 \log g^2] - \mathbb{E}_p[g^2] \log \mathbb{E}_p[g^2] \leq 2C \mathbb{E}_p \|\nabla g\|^2. \tag{2}$$

In the context of Markov diffusions (and in particular, designing sampling algorithms using diffusions), the interest in this quantity comes as it governs the convergence rate of *overdamped* Langevin diffusion in the KL divergence sense. Namely, if $p_t$ is the distribution of overdamped Langevin after time $t$, one can show

$$\mathrm{KL}(p_t\|p) \leq e^{-Ct}\mathrm{KL}(p_0\|p).$$

We will only need the following fact about the log-Sobolev constant:

**Fact 1** (Bakry and Émery [1985], Bakry et al. [2013]). Let the distributions $p(x) \propto \exp(-U(x))$ be such that $U(x) \succeq \lambda I$. Then, $p$ has log-Sobolev constant bounded by $\lambda$.

We will also need the following result characterizing the convergence time of *underdamped* Langevin dynamics in terms of the log-Sobolev constant, as shown in Ma et al. [2019]:

**Theorem 2** (Ma et al. [2019]). *Let $p^*(x) \propto \exp(-U(x))$ have a log-Sobolev constant bounded by $\rho$. Furthermore, for a distribution $p : \mathbb{R}^d \to \mathbb{R}^+$, let*

$$\mathcal{L}[p] := \mathrm{KL}(p\|p^*) + \mathbb{E}_p \left[ \left\langle \nabla \frac{\delta\, \mathrm{KL}(p\|p^*)}{\delta p}, S \nabla \frac{\delta\, \mathrm{KL}(p\|p^*)}{\delta p} \right\rangle \right],$$

*where $S$ is a positive definite matrix given by $S = \frac{1}{\kappa} \begin{bmatrix} \frac{1}{4} I_{d\times d} & \frac{1}{2} I_{d\times d} \\ \frac{1}{2} I_{d\times d} & 2 I_{d\times d} \end{bmatrix}$. If $p_t$ is the distribution of $(x_t, v_t)$ which evolve according to (1), we have*

$$\frac{d}{dt}\mathcal{L}[p_t] \leq -\frac{\rho}{10}\mathcal{L}[p_t] \tag{3}$$

*whenever $p^*$ satisfies a log-Sobolev inequality with constant $\rho$.*

We note that the above theorem uses a non-standard Lyapunov function $\mathcal{L}$, which combines KL divergence with an extra term, since the generator of underdamped Langevin is not self-adjoint—this makes analyzing the drop in $KL$ divergence difficult. As $\mathcal{L}$ is clearly an upper bound on $KL(p\|p^*)$, so it suffices to show $\mathcal{L}$ decreases rapidly.

We will also need a less-well-known *deterministic* form of the updates which is equivalent to (1). Precisely, we convert (1) an equivalent ODE (with time-dependent coefficients). The proof of this fact (via a straightforward comparison of the Fokker-Planck equation) can be found in Ma et al. [2019].

**Theorem 3.** *Let $p_t(x_t, v_t)$ be the probability distribution of running (1) for time $t$. If started from $(x_0, v_0) \sim p_0$, the probability distribution of the solution $(x_t, v_t)$ to the ODEs*

$$\frac{d}{dt} \begin{bmatrix} x_t \\ v_t \end{bmatrix} = \begin{bmatrix} O & I_d \\ -I_d & -\gamma I_d \end{bmatrix} (\nabla \ln p_t - \nabla \ln p^*) \tag{4}$$

*is also $p_t(x_t, v_t)$.*

## 4.2 Dynamical systems and Henon maps

We also build on work from dynamical systems, more precisely, a family of maps called *Hénon-like maps* [Hénon, 1976].

**Definition 8** ([Turaev, 2002])**.** A pair of ODEs forms a *Hénon-like map* if it has the form

$$\begin{cases} \frac{dx}{dt} = v \\ \frac{dv}{dt} = -x + \nabla J(x) \end{cases} \tag{5}$$

for a smooth function $J : \mathbb{R}^d \to \mathbb{R}$.

This special family of ODEs is a continuous-time generalization of a classical discrete dynamical system of the same name Hénon [1976]. The property that is useful for us is that the Euler discretization of this map can be written as a sequence of affine coupling blocks.

In Turaev [2002], it was proven that these ODEs are *universal approximators* in some sense. Namely, the iterations of this ODE can approximate any *symplectic diffeomorphism*: a continuous map which preserves volumes (i.e. the Jacobian of the map is 1). These kinds of diffeomorphisms have their genesis in Hamiltonian formulations of classical mechanics [Abraham and Marsden, 2008].

At first blush, symplectic diffeomorphisms and underdamped Langevin seem to have nothing to do with each other. The connection comes through the so-called Hamiltonian representation theorem [Polterovich, 2012], which states that any symplectic diffeomorphism from $\mathcal{C} \subseteq \mathbb{R}^{2d} \to \mathbb{R}^{2d}$ can be written as the iteration of the following *Hamiltonian* system of ODEs for some (time-dependent) Hamiltonian $H(x, v, t)$:

$$\begin{cases} \frac{dx}{dt} = \frac{d}{dv} H(x, v, t) \\ \frac{dv}{dt} = -\frac{d}{dx} H(x, v, t) \end{cases} \tag{6}$$

In fact, in our theorem, we will use techniques inspired by those in Turaev [2002], who shows:

**Theorem 4** (Turaev [2002])**.** *For any function $H(x, v, t) : \mathbb{R}^{2d} \times \mathbb{R}_{\geq 0} \to \mathbb{R}$ which is polynomial in $(x, v)$, there exists a polynomial $V(x, v, t)$, s.t. the time-$\tau$ map of the system*

$$\begin{cases} \frac{dx}{dt} = \frac{\partial}{\partial v} H(x, v, t) \\ \frac{dv}{dt} = -\frac{\partial}{\partial x} H(x, v, t) \end{cases} \tag{7}$$

*is uniformly $O(\tau^2)$-close to the time-$2\pi$ map of the system*

$$\begin{cases} \frac{dx}{dt} = v \\ \frac{dv_j}{dt} = -\Omega_j^2 x_j - \tau \frac{\partial}{\partial x_j} V(x, t) \end{cases} \tag{8}$$

*for some integers $\{\Omega_i\}_{i=1}^d$.*

We will prove a generalization of this theorem that applies to underdamped Langevin dynamics.

## 5 Proof Sketch of Theorem 1

### 5.1 Overview of strategy

We wish to construct an affine coupling network that (approximately) pushes forward a Gaussian $p^* = \mathcal{N}(0, I_{2d})$ to the distribution we wish to model with Gaussian padding, i.e. $p_0 = p \times \mathcal{N}(0, I_d)$. Because the inverse of an affine coupling network is an affine coupling network, we can invert the problem, and instead attempt to map $p_0$ to $N(0, I_{2d})$. [1]

There is a natural map that takes $p_0$ to $p^* = N(0, I_{2d})$, namely, underdamped Langevin dynamics (1). Hence, our proof strategy involves understanding and simulating underdamped Langevin dynamics with the initial distribution $p_0 = p \times \mathcal{N}(0, I_d)$, and the target distribution $p^* = \mathcal{N}(0, I_{2d})$, and comprises of two important steps.

---

[1]As an aside, a similar strategy is taken in practice by recent SDE-based generative models (Song et al. [2020]).

First, we show that the flow-map for Langevin is well-conditioned (Lemma 1 below). Here, by flow-map, we mean the map which assigns each $x$ to its evolution over a certain amount of time $t$ according to the equations specified by (1).

Second, we break the simulation of underdamped Langevin dynamics for a certain time $t$ into intervals of size $\tau$, and show that the *inverse* flow-map over each $\tau$-sized interval of time can be approximated well by a composition of affine-coupling maps (Lemma 5 below). To show this, we consider a more general system of ODEs than the one in Turaev [2002] (in particular, a non-Hamiltonian system), which can be applied to *underdamped* Langevin dynamics. We then show that the *inverse* flow-map of this system of ODEs can be approximated by a sequence of affine-coupling blocks. We note that for this argument, it is critical that we use underdamped rather than overdamped Langevin dynamics, as overdamped Langevin dynamics do not have the required form for affine-coupling blocks.

## 5.2 Underdamped Langevin is well-conditioned

Consider running underdamped Langevin dynamics with stationary distribution $p^*$ equal to the standard Gaussian, started at a log-concave distribution with bounded condition number $\kappa$. The following lemma says that the flow map is well-conditioned, with condition number depending polynomially on $\kappa$.

**Lemma 1.** *Consider underdamped Langevin dynamics* (1) *with* $\zeta = 1$, *friction coefficient* $\gamma < 2$ *and starting distribution* $p$ *which satisfies all the assumptions in Theorem 1. Let* $T_t$ *denote the flow-map from time* $0$ *to time* $t$ *induced by* (4)*. Then for any* $x_0, v_0 \in \mathbb{R}^d$ *and unit vector* $w$*, the directional derivative of* $T_t$ *at* $x_0, v_0$ *in direction* $w$ *satisfies*

$$\left(1 + \frac{2+\gamma}{2-\gamma}(\kappa - 1)\right)^{-2/\gamma} \leq \|D_w T_t(x_0)\| \leq \left(1 + \frac{2+\gamma}{2-\gamma}(\kappa - 1)\right)^{2/\gamma}.$$

*Therefore, the condition number of* $T_t$ *is bounded by* $\left(1 + \frac{2+\gamma}{2-\gamma}(\kappa - 1)\right)^{4/\gamma}$*.*

We sketch the proof below and include a complete proof in Section A.

First, using (4) and the chain rule shows that the Jacobian of the flow map at $x_0$, $D_t = DT_t(x_0)$, satisfies

$$\frac{d}{dt}D_t = \begin{bmatrix} O & I_d \\ -I_d & -\gamma I_d \end{bmatrix} \nabla^2(\ln p_t - \ln p^*)D_t, \tag{9}$$

i.e., it is bounded by the difference of the Hessians of the log-pdfs of the current distribution and the stationary distribution. We will show that $\nabla^2 \ln p_t$ decays exponentially towards $\nabla^2 \ln p^* = I_{2d}$.

To accomplish this, consider how $\nabla^2 \ln p_t$ evolves if we replace (1) by its discretization,

$$\widetilde{x}_{t+\eta} = \widetilde{x}_t + \eta \widetilde{v}_t$$
$$\widetilde{v}_{t+\eta} = (1 - \eta\gamma)\widetilde{v}_t - \eta\widetilde{x}_t + \xi_t, \quad \xi_t \sim N(0, 2\gamma\eta I_d).$$

Note that because the stationary distribution is a Gaussian, $\nabla U(x_t) = x_t$ in (1), and the above equations take a particularly simple form: we apply a linear transformation to $\begin{bmatrix} \widetilde{x}_t \\ \widetilde{v}_t \end{bmatrix}$, and then add Gaussian noise, which corresponds to convolving the current distribution by a Gaussian. We keep track of upper and lower bounds for $\nabla^2 \ln p_t$, and compute how they evolve under this linear transformation and convolution by a Gaussian. Taking $\eta \to 0$, we obtain differential equations for the upper and lower bounds for $\nabla^2 \ln p_t$, which we can solve. A Grönwall argument shows that these bounds decay exponentially towards $\nabla^2 \ln p^* = I_{2d}$. The decay rate can be bounded as a power of $\frac{1}{\kappa}$.

From (9), we then obtain that the condition number of $D_t$ is bounded by the integral of a exponentially decaying function, and hence is bounded independent of $t$. In particular, we may take $t$ large enough so that $p_t$ is $\epsilon$-close to the stationary distribution. Because the decay rate of the exponential is $\frac{1}{\kappa^{O(1)}}$, the bound is $\kappa^{O(1)}$.

Note that we vitally used the fact that the stationary distribution $p$ is a standard Gaussian, as our argument requires that $\nabla^2 \ln p^*$ be constant everywhere.

## 5.3 ODE approximation by affine-coupling blocks

Next, we analyze a more general version of the Hamiltonian system of ODEs considered in Turaev [2002], which we recalled in (7). In particular, the system of ODEs we will be considering is:

$$\begin{cases} \frac{dx}{dt} = \frac{\partial}{\partial v} H(x, v, t) \\ \frac{dv}{dt} = -\frac{\partial}{\partial x} H(x, v, t) - \gamma \frac{\partial}{\partial v} H(x, v, t) \end{cases} \tag{10}$$

Note that substituting $H(x, v, t) = \ln p_t(x, v) - \ln p^*(x, v)$ above gives us the underdamped Langevin dynamics.

The first step is to restrict our considerations to $H$ being a polynomial in $x, v$, rather than a general smooth function. Towards this, we recall the notion of closeness in the $C^1$ topology:

**Definition 9.** Let $\mathcal{C} \subseteq \mathbb{R}^d$ be a compact set. Let $f, g : \mathcal{C} \to \mathbb{R}$ be two continuously differentiable functions. Then we say that $f, g$ are uniformly $\epsilon$-close over $\mathcal{C}$ in $C^1$ topology if

$$\sup_{x \in \mathcal{C}} (\|f(x) - g(x)\| + \|Df(x) - Dg(x)\|) \leq \epsilon$$

The following lemma (a generalization of the Stone-Weierstrass Theorem) then establishes that it suffices to focus on $H$ being polynomial in $x, v$:

**Lemma 2** (Theorem 5, Peet [2007])**.** *Let $\mathcal{C} \subset \mathbb{R}^d$ be a compact set. For any $C^2$ function $H : \mathbb{R}^d \to \mathbb{R}$, and any $\epsilon > 0$, there is a multivariate polynomial $P : \mathbb{R}^d \to \mathbb{R}$ such that $P, H$ are uniformly $\epsilon$-close over $\mathcal{C}$ in $C^1$ topology.*

Focusing on the case of polynomials, Lemma 3 below shows that instead of flowing the pair of ODEs given by (10) over an interval of time $\tau$, we can instead run a different ODE for time $2\pi$, such that the flow-maps corresponding to both these ODEs are $O(\tau^2)$-close.

**Lemma 3.** *Let $\mathcal{C} \subset \mathbb{R}^{2d}$ be a compact set. For any function $H(x, v, t) : \mathbb{R}^{2d} \to \mathbb{R}$ which is polynomial in $(x, v)$, there exist polynomial functions $J, F, G$, s.t. the time-$(t_0 + \tau, t_0)$ flow map of the system*

$$\begin{cases} \frac{dx}{dt} = \frac{\partial}{\partial v} H(x, v, t) \\ \frac{dv}{dt} = -\frac{\partial}{\partial x} H(x, v, t) - \gamma \frac{\partial}{\partial v} H(x, v, t) \end{cases} \tag{11}$$

*is uniformly $O(\tau^2)$-close over $\mathcal{C}$ in $C^1$ topology to the time-$2\pi$ map of the system*

$$\begin{cases} \frac{dx}{dt} = v - \tau F(v, t) \odot x \\ \frac{dv_j}{dt} = -\Omega_j^2 x_j - \tau J_j(x, t) - \tau v_j G_j(x, t) \end{cases} \tag{12}$$

*Here, $\odot$ denotes component-wise product, and the constants inside the $O(\cdot)$ depend on $\mathcal{C}$ and the coefficients of $H$.*

The complete proof of this lemma is included in Appendix B; we provide a brief sketch here. First, we consider the first order ($O(\tau^2)$) approximation of the flow map of a standard ODE of the form $\dot{y} = Dy$ (where $D$ is diagonal), and observe that for small $\tau$, we can think of (12) as a perturbed version of such an ODE with an appropriate choice of $D$. Using standard ODE perturbation techniques, we can approximately express the time-$t$ evolution of (12) up to first-order in $\tau$, in terms of polynomials $F, G, J$ and trigonometric functions.

Then, we compare this map to the first-order approximation of flowing the pair of ODEs (11) for time $\tau$ via Taylor's theorem. Furthermore, this approximation is a polynomial in $(x, v)$ since $H$ is a polynomial in $(x, v)$.

The crucial step involves choosing the functional form of $F(z, t), J(z, t), G(z, t)$ suitably, so that they are polynomials in $z$ with coefficients in terms of $\sin(\Omega t), \cos(\Omega t)$. After simplification, both expressions can be expressed in terms of polynomials in $x, v$ where coefficients can be expressed in terms of $\int_0^{2\pi} \sin^p(\Omega s) \cos^q(\Omega s)\, ds$, which either integrate to 0 or a constant. Thus, to ensure that the two approximations match, we are left with a problem of making two multivariate polynomials in $(x, v)$ equal.

This final step can of course be written as a linear system of equations. We identify a special structure in this system, which helps us show that the system is full-rank, and hence has a solution. $\qquad \square$

Finally, consider discretizing the newly constructed ODE (12) into small steps of size $\eta$ by a simple Euler schema i.e.,

$$\begin{cases} x_{n+1} = x_n + \eta(v_n - \tau F(v_n, \eta n) \odot x_n) \\ v_{n+1,j} = v_{n,j} - \eta(\Omega_j^2 x_{n,j} - \tau J_j(x_n, \eta n) - \tau v_{n,j} G_j(x_n, \eta n)) \end{cases} \tag{13}$$

We note that each step above can be written as a composition of two affine coupling blocks given by $(x_n, v_n) \mapsto (x_n, v_{n+1}) \mapsto (x_{n+1}, v_{n+1})$. Namely, the map $(x_n, v_n) \mapsto (x_n, v_{n+1})$ can be written as

$$\begin{cases} x_n = x_n \\ v_{n+1} = v_n \odot (1-\tau)G(x_n, \eta n) - \eta(\Omega^2 \odot x_n - \tau J(x_n, \eta n)) \end{cases}$$

This map is an affine coupling block with $s(x_n) = (1-\tau) \odot G(x_n, \eta n)$ and $t(x_n) = -\eta(\Omega^2 \odot x_n - \tau J(x_n, \eta n))$. The map $(x_n, v_{n+1}) \mapsto (x_{n+1}, v_{n+1})$ can be written as

$$\begin{cases} v_{n+1} = v_{n+1} \\ x_{n+1} = x_n + \eta(v_{n+1} - \tau F(v_{n+1}, \eta n) \odot x_n) \end{cases}$$

which is an affine coupling block with $s(v_{n+1}) = 1 - \eta\tau F(v_{n+1}, \eta n)$ and $t(v_{n+1}) = \eta v_{n+1}$.

The composition of the two maps above yields an affine coupling network $(x_n, v_n) \mapsto (x_{n+1}, v_{n+1})$ precisely as given by Equation (13) with non-linearities $s, t$ in each of the blocks given by polynomials. The following lemma bounds the error resulting from this discretization:

**Lemma 4** (Euler's discretization method). [2] *Let $\mathcal{C} \subset \mathbb{R}^{2d}$ be a compact set. Consider discretizing the time from $0$ to $t$ into $\frac{t}{\eta}$ steps and performing the update given by (13) at each of these steps. Let the map obtained as a result of discretizing thus be denoted by $T'_t$ and let the original flow map be denoted by $T_t$. Then $T_t$ and $T'_t$ are uniformly $O(\eta)$ close over $\mathcal{C}$ in $C^1$ topology, and the constants inside the $O(\cdot)$ depend on $\mathcal{C}$, and bounds on the derivatives of $T_t$ over $\mathcal{C}$.*

## 5.4 Simulating by breaking into $\tau$-sized intervals

Let $T_{s,t}$ denote the time-$s, t$ flow-map of (10) from time $s$ to time $t$. Since the flow maps are invertible, $T_{s,t}$ and $T_{t,s}$ are inverses. We are now ready to state the following lemma which says that the underdamped Langevin flow-map $T_{\phi,0}$ can be written as a composition of affine-couplings maps:

**Lemma 5.** *Let $\mathcal{C} \subset \mathbb{R}^{2d}$ be a compact set. Suppose that $T_{\phi,0}(x,v)$ is the time-$(\phi, 0)$ flow-map of the ODE's*

$$\begin{cases} \frac{dx}{dt} = \frac{\partial}{\partial v} H(x,v,t) \\ \frac{dv}{dt} = -\frac{\partial}{\partial x} H(x,v,t) - \gamma \frac{\partial}{\partial v} H(x,v,t) \end{cases} \tag{14}$$

*where $H$ is $C^\infty$. Then for any $\epsilon_1, \phi \in \mathbb{R}_+$, there exists an integer $N = N(\epsilon_1, \phi, \mathcal{C})$ and affine-coupling blocks $f_1, \ldots, f_N$ such that the composition $f = f_N \circ \cdots \circ f_1$ is $\epsilon_1$-close to $T_{\phi,0}$ in the $C^1$ topology over $\mathcal{C}$.*

The proof of Lemma 5 is in Appendix C. We provide a brief sketch here: from Lemma 2, we know that it suffices to show the result for a polynomial $H$. Thereafter, we break the time for which we want to flow the ODE given by (14) into small chunks of length $\tau$. Lemmas 3 and 4 then show that the flow map over this chunk can be written as an affine coupling network. Composing the affine coupling networks over all the chunks of time gives us the result.

## 5.5 Putting components together

The previous sections established that for any $t$ and any compact set $\mathcal{C}$, there is a affine-coupling network $f$ with polynomial non-linearities such that $T_{t,0}$ and $f$ are uniformly close over $\mathcal{C}$. We will now pick an appropriate value of $t$ and set $\mathcal{C}$ such that $W_1(f_\#(p^*|_\mathcal{C}), p_0) \leq \epsilon$ where $p^* = \mathcal{N}(0, \mathrm{I}_{2d})$, which is the required result of Theorem 1. First, using Theorem 2, for

$$\phi > -10 \log \epsilon_1 + \log 2 + \log \mathcal{L}[p_0]$$

we have that $\mathrm{KL}(T_{0,\phi\#}(p_0), p^*) \leq \frac{\epsilon_1^2}{2}$. We use the following *transportion cost inequality* to convert this to a Wasserstein bound.

---

[2]This result is well known in the $C^0$ topology, we provide an analysis for the $C^1$ bound in Appendix D.1.

**Theorem 5** (Talagrand [1996])**.** *The standard Gaussian $p$ on $\mathbb{R}^d$ satisfies a transportation cost inequality: For every distribution $q$ on $\mathbb{R}^d$ with finite second moment, $W_1(p,q)^2 \leq 2KL(q\|p)$.*

This gives us that $W_1(T_{0,\phi\#}(p_0), p^*) \leq \epsilon_1$. A simple argument in Lemma 10 (Appendix D.2) then gives

$$W_1(p_0, T_{\phi,0\#}(p^*)) = W_1(T_{\phi,0\#}(T_{0,\phi\#}(p_0)), T_{\phi,0\#}(p^*)) \leq \mathrm{Lip}(T_{\phi,0})\epsilon_1 \tag{15}$$

A subsequent argument stated as Lemma 11 in Appendix D.2, shows that if $f$ and $T_{\phi,0}$ are uniformly $\epsilon_1$-close in $C^0$ topology on some $\mathcal{C}$, then their pushforwards through $p^*|_{\mathcal{C}}$ are indeed close, i.e.,

$$W_1(T_{\phi,0\#}(p^*|_{\mathcal{C}}), f_\#(p^*|_{\mathcal{C}})) \leq \epsilon_1. \tag{16}$$

Next, we establish a bound on the Wasserstein distance between the standard Gaussian and its truncation on a compact set, proved in Appendix D.3.

**Lemma 6.** *Let $p^* = \mathcal{N}(0, \mathrm{I}_{2d})$. Then for every $\delta \in \mathbb{R}_+$, there exists a compact set $\mathcal{C} = B(0,R)$ such that $W_1(p^*, p^*|_{\mathcal{C}}) \leq \delta$, where $B(0,R)$ denotes the ball of radius $R$ centered at the origin.*

We now choose a compact set $\mathcal{C}$ such that Lemma 6 holds for $\delta = \epsilon_1$. Then Lemma 10 again implies that

$$W_1(T_{\phi,0\#}(p^*), T_{\phi,0\#}(p^*|_{\mathcal{C}})) \leq \mathrm{Lip}(T_{\phi,0})\epsilon_1 \tag{17}$$

Equations (15), (16), (17) and the triangle inequality together imply

$$W_1(f_\#(p^*|_{\mathcal{C}}), p_0) \leq (2\mathrm{Lip}(T_{\phi,0}) + 1)\epsilon_1 \leq \epsilon$$

for small enough $\epsilon_1$. We can indeed set $\epsilon_1$ small enough so as to satisfy the last inequality above, because of the global bound $\mathrm{Lip}(T_{\phi,0}) \leq \left(1 + \frac{2+\gamma}{2-\gamma}(\kappa - 1)\right)^{2/\gamma}$ established in Lemma 1. This gives us the statement of Theorem 1. Note that the final value of $\phi$ depends on $\epsilon, \kappa, \gamma$ and $\mathcal{L}[p_0]$.

# 6 Conclusion

In this paper, we provide the first guarantees on universal approximation with *well-conditioned* affine coupling networks. The conditioning of the network is crucial when the networks are trained using gradient-based optimization of the likelihood. Mathematically, we uncover connections between stochastic differential equations, dynamical systems and affine coupling flows. Our construction uses Gaussian padding, which lends support to the empirical observation that this strategy tends to result in better-conditioned flows [Koehler et al., 2020]. We leave it as an open problem to generalize beyond log-concave distributions.

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
