## A  Conditioning

We analyze the condition number of underdamped Langevin dynamics with potential $f(x) = \frac{1}{2}\|x\|^2$ and stationary distribution $p(x,v) = e^{-f(x)-\frac{1}{2}\|v\|^2} = e^{-\frac{1}{2}(\|x\|^2+\|v\|^2)}$. Underdamped Langevin dynamics is given by the following SDE's,

$$dx_t = -v_t \tag{18}$$

$$dv_t = -\gamma v_t - \nabla f(x_t) + \sqrt{2}dB_t$$

$$= -\gamma v_t - x_t + \sqrt{2}dB_t. \tag{19}$$

Given the distribution $p_0$ at time 0, the distribution $p_t$ at time $t$ is the same as that given by,

$$\begin{bmatrix} \frac{dx}{dt} \\ \frac{dv}{dt} \end{bmatrix} = - \begin{bmatrix} 0 & -\mathrm{I}_d \\ \mathrm{I}_d & \gamma\mathrm{I}_d \end{bmatrix} \begin{bmatrix} \nabla_x \frac{\delta \, \mathrm{KL}(\mathbf{p}_t\|\mathbf{p}^*)}{\delta \mathbf{p}_t} \\ \nabla_v \frac{\delta \, \mathrm{KL}(\mathbf{p}_t\|\mathbf{p}^*)}{\delta \mathbf{p}_t} \end{bmatrix} \tag{20}$$

which simplifies to

$$d \begin{bmatrix} x_t \\ v_t \end{bmatrix} = \begin{bmatrix} O & \mathrm{I}_d \\ -\mathrm{I}_d & -\gamma\mathrm{I}_d \end{bmatrix} (\nabla \ln p_t - \nabla \ln p). \tag{21}$$

Our goal is to prove the following theorem.

**Theorem 6.** *Consider underdamped Langevin dynamics* (18)–(19) *with friction coefficient* $\gamma < 2$ *and starting distribution* $p_0$ *that is* $C^2$. *Let* $T_t$ *denote the transport map from time 0 to time* $t$ *induced by* (21). *Suppose that the initial distribution* $p_0(x,v)$ *is such that*

$$\mathrm{I}_{2d} \preceq -\nabla^2 \ln p_0(x,v) \preceq \kappa \mathrm{I}_{2d}.$$

*Then for any* $x_0, v_0$ *and unit vector* $w$, *the directional derivative of* $T_t$ *at* $x_0, v_0$ *in direction* $w$ *satisfies*

$$\left(1 + \frac{2+\gamma}{2-\gamma}(\kappa-1)\right)^{-2/\gamma} \leq \|D_w T_t(x_0)\| \leq \left(1 + \frac{2+\gamma}{2-\gamma}(\kappa-1)\right)^{2/\gamma}$$

*Thus the condition number of* $T_t$ *is bounded by* $\left(1 + \frac{2+\gamma}{2-\gamma}(\kappa-1)\right)^{4/\gamma}$.

We remark that the exponent is likely loose by a factor of 2, and that taking $\gamma \to 2$ gives the best exponent; however, the case $\gamma = 2$ would require a separate calculation as the matrix appearing in the exponential is not diagonalizable. Note $\gamma = 2$ is the transition between when the dynamics exhibit underdamped and overdamped behavior.

To prove the theorem, we first relate the Jacobian with the Hessian of the log-pdf. By Lemma 12, the Jacobian $D_t = DT_t(x_0)$ satisfies

$$\frac{d}{dt} D_t = \begin{bmatrix} O & \mathrm{I}_d \\ -\mathrm{I}_d & -\gamma\mathrm{I}_d \end{bmatrix} \nabla^2 (\ln p_t - \ln p) D_t. \tag{22}$$

We will show that $\nabla^2(\ln p_t - \ln p)$ decays exponentially (Lemma 8). First, we need the following bound for convolutions.

### A.1  Bounding the Hessian of the logarithm of a convolution

**Lemma 7.** *Suppose that* $p$ *is a probability density function on* $\mathbb{R}^d$ *such that* $\Sigma_1^{-1} \preceq -\nabla^2 \ln p \preceq \Sigma_2^{-1}$. *Let* $q$ *be the distribution of* $N(0,\Sigma)$ *(where* $\Sigma$ *is not necessarily full-rank). Then*

$$(\Sigma_1 + \Sigma)^{-1} \preceq -\nabla^2 \ln(p*q) \preceq (\Sigma_2 + \Sigma)^{-1}.$$

*Proof.* The lower bound is a bound on the strong log-concavity parameter; see Theorem 3.7b in Saumard and Wellner [2014].

For the upper bound, we first prove the lemma in the case that $\Sigma$ is full rank. We have $(p * q)(x) = \int_{\mathbb{R}^d} p(u)q(x-u)\, dt$, so

$$\nabla^2[\ln((p*q)(x))] = \frac{\int_{\mathbb{R}^d} p(u)\nabla^2 q(x-u)\, du}{\int_{\mathbb{R}^d} p(u)q(x-u)\, du} - \left(\frac{\int_{\mathbb{R}^d} p(u)\nabla q(x-u)\, du}{\int_{\mathbb{R}^d} p(u)q(x-u)\, du}\right)\left(\frac{\int_{\mathbb{R}^d} p(u)\nabla q(x-u)\, du}{\int_{\mathbb{R}^d} p(u)q(x-u)\, du}\right)^\top$$

$$= \left(\frac{\int_{\mathbb{R}^d} \Sigma^{-1}(x-u)p(u)q(x-u)\, du}{\int_{\mathbb{R}^d} p(u)q(x-u)\, du}\right)\left(\frac{\int_{\mathbb{R}^d} (\Sigma^{-1}(x-u))^\top p(u)q(x-u)\, du}{\int_{\mathbb{R}^d} p(u)q(x-u)\, du}\right)$$

$$- \frac{\int_{\mathbb{R}^d} (\Sigma^{-1}(x-u)(x-u)^\top \Sigma^{-1} - \Sigma^{-1})p(u)q(x-u)\, du}{\int_{\mathbb{R}^d} p(u)q(x-u)\, du}$$

Let $\mu_x$ denote the distribution with density function $\rho(u) \propto p(u)q(x-u)$. Then

$$-\nabla^2[\ln((p*q)(x))] = [\mathbb{E}_{\mu_x}\Sigma^{-1}(u-x)][\mathbb{E}_{\mu_x}(\Sigma^{-1}(u-x))^\top] - [\mathbb{E}_{\mu_x}\Sigma^{-1}(u-x)(u-x)^\top \Sigma^{-1}] + \Sigma^{-1}$$

$$= -\mathbb{E}_{\mu_x}[\Sigma^{-1}(u-\mathbb{E}u)(u-\mathbb{E}u)^\top \Sigma^{-1}] + \Sigma^{-1}.$$

It suffices to show for any unit vector $v$, that

$$-v^\top \nabla^2[\ln((p*q)(x))]v = -\mathbb{E}_{\mu_x}[\langle \Sigma^{-1}v, (u-\mathbb{E}u)\rangle^2] + v^\top \Sigma^{-1}v \leq v^\top(\Sigma_2 + \Sigma)^{-1}v$$

Note that $\mu_x$ satisfies

$$-\nabla^2 \ln \mu_x \preceq \Sigma_2^{-1} + \Sigma^{-1},$$

so $\mu_x$ can be written as the density of a Gaussian with variance $(\Sigma_2^{-1} + \Sigma^{-1})^{-1}$ multiplied by a log-convex function. By the Brascamp-Lieb moment inequality (Theorem 5.1 in Brascamp and Lieb [2002])[3],

$$\mathbb{E}_{\mu_x}[\langle \Sigma^{-1}v, (u-\mathbb{E}u)\rangle^2] \geq \mathbb{E}_{u \sim N(0,(\Sigma_2^{-1}+\Sigma^{-1})^{-1})}[\langle \Sigma^{-1}v, u\rangle^2] = v^\top \Sigma^{-1}(\Sigma_2^{-1} + \Sigma^{-1})^{-1}\Sigma^{-1}v.$$

Hence

$$-v^\top \nabla^2[\ln((p*q)(x))]v \leq v^\top \left[-\Sigma^{-1}(\Sigma_2^{-1} + \Sigma^{-1})^{-1}\Sigma^{-1} + \Sigma^{-1}\right]v$$

The conclusion then follows from

$$-\Sigma^{-1}(\Sigma_2^{-1} + \Sigma^{-1})^{-1}\Sigma^{-1} + \Sigma^{-1} = -(\Sigma\Sigma_2^{-1}\Sigma + \Sigma)^{-1} + \Sigma^{-1}$$

$$= (\Sigma\Sigma_2^{-1}\Sigma + \Sigma)^{-1}(-\cancel{I_d} + \Sigma\Sigma_2^{-1} + \cancel{I_d})$$

$$= (\Sigma + \Sigma_2)^{-1}.$$

Now for the general case, take the limit as $\Sigma' \to \Sigma$ where $\Sigma'$ is full-rank. More precisely, let $\Sigma_t = \Sigma + tP$, where $P$ is projection onto $\mathrm{Im}(\Sigma)^\perp$, and let $q_t$ be the density function for $N(0, \Sigma_t)$. Then we have

$$\nabla^2[\ln((p*q_t)(x))] = \frac{\int_{\mathbb{R}^d} \nabla^2 p(x-u)q_t(u)\, du}{\int_{\mathbb{R}^d} p(x-u)q_t(u)\, du} - \left(\frac{\int_{\mathbb{R}^d} \nabla p(x-u)q_t(u)\, du}{\int_{\mathbb{R}^d} p(x-u)q_t(u)\, du}\right)\left(\frac{\int_{\mathbb{R}^d} \nabla p(x-u)q_t(u)\, du}{\int_{\mathbb{R}^d} p(x-u)q_t(u)\, du}\right)^\top$$

Examining the first term, we have

$$\int_{\mathbb{R}^d} \nabla^2 p(x-u)q_t(u)\, du = \int_{\mathrm{Im}(\Sigma)}\int_{\mathrm{Im}(P)} \nabla^2 p(x-u-v)q_t(u+v)\, dv\, du$$

$$\to \int_{\mathrm{Im}(\Sigma)} \nabla^2 p(x-u)q_t(u)\, du \text{ as } t \to 0^+$$

by the dominated convergence theorem. Similarly, the other integrals converge to their counterparts with $q(u)$. Therefore, $\nabla^2[\ln((p*q_t)(x))] \to \nabla^2[\ln((p*q)(x))]$ as $t \to 0^+$. Apply the lemma to the full-rank case; the RHS bound converges to the desired bound: $(\Sigma_2 + \Sigma_t)^{-1} \to (\Sigma_2 + \Sigma)^{-1}$.

$\square$

---

[3]Note that the sign is flipped in the theorem statement in the log-convex case.

## A.2 Bounding the variance proxy for underdamped Langevin

As it is useful to work with the matrices $\Sigma_1$ and $\Sigma_2$, we make the following definition.

**Definition 10.** Let $p$ be a probability density on $\mathbb{R}^d$. For a positive definite matrix $\Sigma_1$, if $\Sigma_1^{-1} \preceq -\nabla^2 \ln p$, we say that $\Sigma_1$ is an **upper variance proxy** for $p$. For a positive definite matrix $\Sigma_2$, if $-\nabla^2 \ln p \preceq \Sigma_2^{-1}$, we say $\Sigma_2$ is a **lower variance proxy** for $p$.

**Lemma 8.** *Consider underdamped Langevin dynamics* (18)–(19) *with with starting distribution $p_0(x, v)$ that is $C^2$. Suppose $p_0$ has lower (upper) variance proxy $\Sigma_0$. Then $p_t$ has lower (upper) variance proxy*

$$\Sigma_t = \exp\left[\left(\begin{bmatrix} & 1 \\ -1 & -\gamma \end{bmatrix} \otimes \mathrm{I}_d\right) t\right] (\Sigma_0 - \mathrm{I}_{2d}) \exp\left[\left(\begin{bmatrix} & -1 \\ 1 & -\gamma \end{bmatrix} \otimes \mathrm{I}_d\right) t\right] + \mathrm{I}_{2d}.$$

*Proof.* We first consider discretized Lanegvin, given by

$$\widetilde{x}_{t+\eta} = \widetilde{x}_t + \eta \widetilde{v}_t$$
$$\widetilde{v}_{t+\eta} = (1 - \eta\gamma)\widetilde{v}_t - \eta\widetilde{x}_t + \xi_t, \quad \xi_t \sim N(0, 2\eta \mathrm{I}_d)$$

or in matrix form,

$$\begin{bmatrix} \widetilde{x}_{t+\eta} \\ \widetilde{v}_{t+\eta} \end{bmatrix} = \begin{bmatrix} \mathrm{I}_d & \eta\mathrm{I}_d \\ -\eta\mathrm{I}_d & (1 - \eta\gamma)\mathrm{I}_d \end{bmatrix} \begin{bmatrix} \widetilde{x}_t \\ \widetilde{v}_t \end{bmatrix} + \xi_t, \quad \xi_t \sim N\left(0, \begin{bmatrix} O & O \\ O & 2\eta\mathrm{I}_d \end{bmatrix}\right).$$

Fix $t$. Let $\widetilde{p}_t^{(\eta)}$ be the distribution at time $t$ for discretized Langevin with step size $\eta$ (dividing $t$). By standard arguments, $\widetilde{p}_t^{(\eta)} \to p_t$ as $\eta \to 0$, in the $C^2$ topology on any compact set. In particular, for any $x, v$, $\nabla^2 \ln \widetilde{p}_t^{(\eta)}(x, v) \to \nabla^2 \ln p_t(x, v)$. Hence it suffices to bound $\nabla^2 \ln p_t(x, v)$.

We write the proof for the upper variance proxy; the proof for the lower variance proxy differs only in the direction of the inequality. Suppose $-\ln \widetilde{p}_t(x, v) \succeq \widetilde{\Sigma}_t^{-1}$. Consider breaking the update into two steps,

$$\begin{bmatrix} \widetilde{x}'_{t+\eta} \\ \widetilde{v}'_{t+\eta} \end{bmatrix} = \begin{bmatrix} \mathrm{I}_d & \eta\mathrm{I}_d \\ -\eta\mathrm{I}_d & (1 - \eta\gamma)\mathrm{I}_d \end{bmatrix} \begin{bmatrix} \widetilde{x}_t \\ \widetilde{v}_t \end{bmatrix}$$
$$\begin{bmatrix} \widetilde{x}_{t+\eta} \\ \widetilde{v}_{t+\eta} \end{bmatrix} = \begin{bmatrix} \widetilde{x}'_{t+\eta} \\ \widetilde{v}'_{t+\eta} \end{bmatrix} + \xi_t, \quad \xi_t \sim N\left(0, \begin{bmatrix} O & O \\ O & 2\eta\mathrm{I}_d \end{bmatrix}\right).$$

Let $\widetilde{p}'_{t+\eta}(x, v)$ denote the distribution of $\begin{bmatrix} \widetilde{x}'_{t+\eta} \\ \widetilde{v}'_{t+\eta} \end{bmatrix}$. Then

$$\widetilde{p}'_{t+\eta}(x, v) = \widetilde{p}_t\left(\begin{bmatrix} \mathrm{I}_d & \eta\mathrm{I}_d \\ -\eta\mathrm{I}_d & (1 - \eta\gamma)\mathrm{I}_d \end{bmatrix}^{-1} \begin{bmatrix} x \\ v \end{bmatrix}\right)$$

so

$$\widetilde{\Sigma}'_{t+\eta} := \begin{bmatrix} \mathrm{I}_d & \eta\mathrm{I}_d \\ -\eta\mathrm{I}_d & (1 - \eta\gamma)\mathrm{I}_d \end{bmatrix} \widetilde{\Sigma}_t \begin{bmatrix} \mathrm{I}_d & -\eta\mathrm{I}_d \\ \eta\mathrm{I}_d & (1 - \eta\gamma)\mathrm{I}_d \end{bmatrix}$$

is an upper variance proxy for $\widetilde{p}'_{t+\eta}$ and by Lemma 7,

$$\widetilde{\Sigma}_{t+\eta} := \widetilde{\Sigma}'_{t+\eta} + \begin{bmatrix} O & O \\ O & 2\eta\mathrm{I}_d \end{bmatrix}$$

is an upper variance proxy for $\widetilde{p}_{t+\eta}$. Note that

$$\widetilde{\Sigma}_{t+\eta} := \widetilde{\Sigma}_t + \left[\begin{bmatrix} & 1 \\ -1 & -\gamma\eta \end{bmatrix} \otimes \mathrm{I}_d\right] \widetilde{S}_t + \widetilde{S}_t\left[\begin{bmatrix} & -1 \\ 1 & -\gamma\eta \end{bmatrix} \otimes \mathrm{I}_d\right] + \begin{bmatrix} 0 & 0 \\ 0 & 2\gamma\eta \end{bmatrix} + O(\eta^2).$$

By the standard analysis of Euler's method, as $\eta \to 0$, the distribution, $\widetilde{\Sigma}_t$ approaches $\Sigma_t$ defined by

$$\frac{d}{dt}\Sigma_t = \left[\begin{bmatrix} & 1 \\ -1 & -\gamma \end{bmatrix} \otimes \mathrm{I}_d\right] \Sigma_t + \Sigma_t\left[\begin{bmatrix} & -1 \\ 1 & -\gamma \end{bmatrix} \otimes \mathrm{I}_d\right] + \begin{bmatrix} 0 & 0 \\ 0 & 2\gamma \end{bmatrix}.$$

This $\Sigma_t$ is an upper variance proxy for $p_t$. The solution to this equation is

$$\Sigma_t = \exp\left[\left(\begin{bmatrix} & 1 \\ -1 & -\gamma \end{bmatrix} \otimes \mathrm{I}_d\right) t\right] (\Sigma_0 - \mathrm{I}_{2d}) \exp\left[\left(\begin{bmatrix} & -1 \\ 1 & -\gamma \end{bmatrix} \otimes \mathrm{I}_d\right) t\right] + \mathrm{I}_{2d},$$

as desired. $\qquad\square$

### A.3 Proof that underdamped Langevin is well-conditioned

We are now ready to prove the main theorem.

*Proof of Theorem 6.* Let $H_t = \nabla^2(-\ln p_t + \ln p)$ and $C = \begin{bmatrix} O & \mathrm{I}_d \\ -\mathrm{I}_d & -\gamma\mathrm{I}_d \end{bmatrix}$. By (22) and the chain rule,

$$\frac{d}{dt}D_tD_t^\top = -(CH_tD_tD_t^\top + D_tD_t^\top H_tC^\top). \tag{23}$$

Fix $w$ and consider $y_t = D_t w = D_w T_t(x_0)$. Multiplying the above by $W$ on both sides gives[4]

$$\left|\frac{d}{dt}\|y_t\|^2\right| \le 2\|CH_t\|\,\|y_t\|^2$$

so by Grönwall's inequality (Lemma 15),

$$\exp\left[-2\int_0^t \|CH_s\|\,ds\right] \le \|y_t\|^2 \le \exp\left[2\int_0^t \|CH_s\|\,ds\right]. \tag{24}$$

By Lemma 8,

$$\mathrm{I}_{2d} \preceq -\nabla^2\ln p_t \preceq (\kappa-1)\exp\left[\left(\begin{bmatrix} -1 & 1 \\ -1 & -\gamma \end{bmatrix}\otimes\mathrm{I}_d\right)t\right]\exp\left[\left(\begin{bmatrix} 1 & -1 \\ 1 & -\gamma \end{bmatrix}\otimes\mathrm{I}_d\right)t\right] + \mathrm{I}_{2d}.$$

The eigenvalues of $A := \begin{bmatrix} 1 & -1 \\ 1 & -\gamma \end{bmatrix}$ are $\frac{-\gamma\pm\sqrt{\gamma^2-4}}{2}$, which have absolute value 1. The absolute value of the inner product of the eigenvectors of $A$ is $\gamma/2$, so the condition number squared of the two exponential factors is bounded by $\frac{1+\frac{\gamma}{2}}{1-\frac{\gamma}{2}} = \frac{2+\gamma}{2-\gamma}$. In full detail, we calculate

$$\exp\left(\begin{bmatrix} 1 & -1 \\ 1 & \gamma \end{bmatrix}t\right) = \underbrace{\begin{bmatrix} 1 & 1 \\ \frac{\gamma-\sqrt{\gamma^2-4}}{2} & \frac{\gamma+\sqrt{\gamma^2-4}}{2} \end{bmatrix}}_{S}\underbrace{\begin{bmatrix} \exp\left(\frac{-\gamma+\sqrt{\gamma^2-4}}{2}t\right) & \\ & \exp\left(\frac{-\gamma-\sqrt{\gamma^2-4}}{2}t\right) \end{bmatrix}}_{D}$$

$$\cdot\underbrace{\frac{1}{\sqrt{\gamma^2-4}}\begin{bmatrix} \frac{\gamma+\sqrt{\gamma^2-4}}{2} & -1 \\ \frac{-\gamma+\sqrt{\gamma^2-4}}{2} & 1 \end{bmatrix}}_{S^{-1}}$$

$$\|S^\dagger S\| = \left\|\begin{bmatrix} 2 & \frac{\gamma^2+\gamma\sqrt{\gamma^2-4}}{2} \\ \frac{\gamma^2-\gamma\sqrt{\gamma^2-4}}{2} & 2 \end{bmatrix}\right\| = 2+\gamma$$

$$\left\|\exp\left(\begin{bmatrix} 1 & -1 \\ 1 & \gamma \end{bmatrix}t\right)\right\| \le \frac{2+\gamma}{\sqrt{4-\gamma^2}}\exp\left(\frac{-\gamma t}{2}\right) = \sqrt{\frac{2+\gamma}{2-\gamma}}\exp\left(\frac{-\gamma t}{2}\right).$$

Hence $H_t = -\nabla^2\ln p_t + \mathrm{I}_{2d}$ satisfies

$$\|CH_s\| \le 1 - \frac{1}{1+\frac{2+\gamma}{2-\gamma}(\kappa-1)e^{-\gamma t/2}}$$

$$\int_0^\infty \|CH_s\|\,ds \le \int_0^\infty \frac{\frac{2+\gamma}{2-\gamma}(\kappa-1)e^{-\gamma t/2}}{1+\frac{2+\gamma}{2-\gamma}(\kappa-1)e^{-\gamma t/2}}\,ds$$

$$\le \left[\frac{2}{\gamma}\ln\left(1+\frac{2+\gamma}{2-\gamma}(\kappa-1)e^{-\gamma t/2}\right)\right]_\infty^0 \le \frac{2}{\gamma}\ln\left(1+\frac{2+\gamma}{2-\gamma}(\kappa-1)\right).$$

---

[4]The condition number bound in Theorem 6 is the square of what one might expect because we are only able to get obtain a bound on the absolute value here. If this is always increasing or decreasing, then we would save a factor of 2 in the exponent.

Hence by (24),

$$\left(1 + \frac{2+\gamma}{2-\gamma}(\kappa - 1)\right)^{-2/\gamma} \leq \|y_t\| \leq \left(1 + \frac{2+\gamma}{2-\gamma}(\kappa - 1)\right)^{2/\gamma},$$

giving the theorem. To obtain the bound on condition number, note that the condition number of $DT_t(x_0)$ is $\frac{\max_{\|w\|=1}\|D_wT_t(x_0)\|}{\min_{\|w\|=1}\|D_wT_t(x_0)\|}$. $\qquad\square$

# B   Proof of Lemma 3

For the sake of convenience, we restate Lemma 3 again.

**Lemma.** *Let $\mathcal{C} \in \mathbb{R}^{2d}$ be a compact set. For any function $H(x,v,t) : \mathbb{R}^{2d} \times \mathbb{R}_{\geq 0} \to \mathbb{R}$ which is polynomial in $(x,v)$, there exist polynomial functions $J$, $F$, $G$, s.t. the time-$(t_0 + \tau, t_0)$ flow map of the system*

$$\begin{cases} \frac{dx}{dt} = \frac{\partial}{\partial v}H(x,v,t) \\ \frac{dv}{dt} = -\frac{\partial}{\partial x}H(x,v,t) - \gamma\frac{\partial}{\partial v}H(x,v,t) \end{cases} \tag{25}$$

*is uniformly $O(\tau^2)$-close over $\mathcal{C}$ in $C^1$ topology to the time-$2\pi$ map of the system*

$$\begin{cases} \frac{dx}{dt} = v - \tau F(v,t) \odot x \\ \frac{dv_j}{dt} = -\Omega_j^2 x_j - \tau J_j(x,t) - \tau v_j G_j(x,t) \end{cases} \tag{26}$$

*for some integers $\{\Omega_j\}_{j=1}^d$. Here, $\odot$ denotes component-wise product, and the constants inside the $O(\cdot)$ depend on $\mathcal{C}$ and the coefficients of $H$.*

*Proof.* First, note that the time-$(t_0 + \tau, t_0)$ flow map of (25) is equal to the time-$(t_0, t_0 + \tau)$ flow map of the system:

$$\begin{cases} \frac{dx}{dt} = -\frac{\partial}{\partial v}H(x,v,t_0 + \tau - t) \\ \frac{dv}{dt} = \frac{\partial}{\partial x}H(x,v,t_0 + \tau - t) + \gamma\frac{\partial}{\partial v}H(x,v,t_0 + \tau - t) \end{cases} \tag{27}$$

Proceeding ahead, we broadly follow the proof strategy in Turaev [2002]. For notational convenience, let's denote the initial vector by $x(0), v(0)$ (each coordinate is specified separately). Let

$$x_j^0(t) = x_j(0)\cos\Omega_j t + \frac{1}{\Omega_j}v_j(0)\sin\Omega_j t \tag{28}$$

$$v_j^0(t) = -\Omega_j x_j(0)\sin\Omega_j t + v_j(0)\cos\Omega_j t. \tag{29}$$

Using perturbative ODE techniques (see appendix D.5), the solution to (26) satisfies

$$\begin{cases} x(t) = x^0(t) - \tau\int_0^t \left(\frac{1}{\Omega}\odot J(x^0(s),s)\odot\sin\Omega(t-s) + F(v^0(s),s)\odot\cos\Omega(t-s)\odot x^0(s)\right. \\ \qquad\qquad \left. + \frac{1}{\Omega}\odot G(x^0(s),s)\odot\sin\Omega(t-s)\odot v^0(s)\right)ds + O(\tau^2) \\ v(t) = v^0(t) - \tau\int_0^t \left(J(x^0(s),s)\odot\cos\Omega(t-s) - \Omega\odot F(v^0(s),s)\odot\sin\Omega(t-s)\odot x^0(s)\right. \\ \qquad\qquad \left. + G(x^0(s),s)\odot\cos\Omega(t-s)\odot v^0(s)\right)ds + O(\tau^2) \end{cases} \tag{30}$$

Substituting $t = 2\pi$, the time-$2\pi$ map of (26) is given by

$$\begin{cases} x(2\pi) = x^0(2\pi) - \tau\int_0^{2\pi}\left(-\frac{1}{\Omega}\odot J(x^0(s),s)\odot\sin\Omega s + F(v^0(s),s)\odot\cos\Omega s\odot x^0(s)\right. \\ \qquad\qquad \left. -\frac{1}{\Omega}\odot G(x^0(s),s)\odot\sin\Omega s\odot v^0(s)\right)ds + O(\tau^2) \\ v(2\pi) = v^0(2\pi) - \tau\int_0^{2\pi}\left(J(x^0(s),s)\odot\cos\Omega s + \Omega\odot F(v^0(s),s)\odot\sin\Omega s\odot x^0(s)\right. \\ \qquad\qquad \left. + G(x^0(s),s)\odot\cos\Omega s\odot v^0(s)\right)ds + O(\tau^2) \end{cases} \tag{31}$$

Note that this holds if $\Omega$ is integral, and we will choose it to be so.

On the other hand, using Taylor's theorem, the solution to (25) satisfies:

$$\begin{cases} x(\tau) = x(0) - \tau\frac{\partial}{\partial v}H(x(0),v(0),t_0 + \tau) + O(\tau^2) \\ v(\tau) = v(0) + \tau\frac{\partial}{\partial x}H(x(0),v(0),t_0 + \tau) + \tau\gamma\frac{\partial}{\partial v}H(x(0),v(0),t_0 + \tau) + O(\tau^2) \end{cases} \tag{32}$$

We will now show that for any two polynomials $r_1, r_2$ of total degree at most $M$ we can choose functions $J, F, G$, s.t.:

$$\begin{cases} \int_0^{2\pi} \left( -\frac{1}{\Omega} \odot J(x^0(s), s) \odot \sin \Omega s + F(v^0(s), s) \odot \cos \Omega s \odot x^0(s) \right. \\ \qquad\qquad \left. -\frac{1}{\Omega} \odot G(x^0(s), s) \odot \sin \Omega s \odot v^0(s) \right) ds = r_1(x(0), y(0)) \\ \int_0^{2\pi} \left( J(x^0(s), s) \odot \cos \Omega s + \Omega \odot F(v^0(s), s) \odot \sin \Omega s \odot x^0(s) \right. \\ \qquad\qquad \left. + G(x^0(s), s) \odot \cos \Omega s \odot v^0(s) \right) ds = r_2(x(0), y(0)) \end{cases} \tag{33}$$

We will choose $J, F, G$ of the form:

$$\begin{cases} \forall j \in [d] : J_j(z, t) = \sum_{\mathbf{i}:|\mathbf{i}| \le M} v_{j,\mathbf{i}}^J(t) z^{\mathbf{i}} \\ \forall j \in [d] : F_j(z, t) = \sum_{\mathbf{i}:|\mathbf{i}| \le M-1} v_{j,\mathbf{i}}^F(t) z^{\mathbf{i}} \\ \forall j \in [d] : G_j(z, t) = \sum_{\mathbf{i}:|\mathbf{i}| \le M-1} v_{j,\mathbf{i}}^G(t) z^{\mathbf{i}} \end{cases} \tag{34}$$

where $\mathbf{i} = (i_1, \dots, i_d)$ denotes multi-index, and $|\mathbf{i}| = \sum_{k=1}^d i_k$ and $z^{\mathbf{i}} = \prod_{k=1}^d z_k^{i_k}$. Let

$$r_{1,j}(x(0), v(0)) = \sum_{\mathbf{k}:|\mathbf{k}| \le M} \sum_{\mathbf{p}+\mathbf{q}=\mathbf{k}} h_{j,\mathbf{p},\mathbf{q}}^1 x(0)^{\mathbf{p}} v(0)^{\mathbf{q}} \tag{35}$$

$$r_{2,j}(x(0), v(0)) = \sum_{\mathbf{k}:|\mathbf{k}| \le M} \sum_{\mathbf{p}+\mathbf{q}=\mathbf{k}} h_{j,\mathbf{p},\mathbf{q}}^2 x(0)^{\mathbf{p}} v(0)^{\mathbf{q}} \tag{36}$$

The equation (33) gives us that for all $j$,

$$\begin{cases} \int_0^{2\pi} \left( -\frac{1}{\Omega_j} J_j(x^0(s), s) \sin(\Omega_j s) + F_j(v^0(s), s) \cos(\Omega_j s) x_j^0(s) \right. \\ \qquad\qquad \left. -\frac{1}{\Omega_j} G_j(x^0(s), s) \sin(\Omega_j s) v_j^0(s) \right) ds = r_{1,j}(x(0), y(0)) \\ \int_0^{2\pi} \left( J_j(x^0(s), s) \cos(\Omega_j s) + \Omega_j F_j(v^0(s), s) \sin(\Omega_j s) x_j^0(s) \right. \\ \qquad\qquad \left. + G_j(x^0(s), s) \cos(\Omega_j s) v_j^0(s) \right) ds = r_{2,j}(x(0), y(0)) \end{cases} \tag{37}$$

Let $\binom{\mathbf{k}}{\mathbf{p}} = \prod_{k=1}^d \binom{k_i}{p_i}$. Let $\mathbf{k_j^t}$ be the multi-index $(k_1, \dots, k_j + t, \dots, k_d)$. We substitute (28)–(29), (34), and (35)–(36) into (37) and match the coefficients of $x(0)^{\mathbf{p}} v(0)^{\mathbf{q}}$.

If $k_j = 0$, then

$$h_{j,\mathbf{p},\mathbf{q}}^1 = \int_0^{2\pi} -\frac{1}{\Omega_j} v_{j,\mathbf{k}}^J \cos(\Omega s)^{\mathbf{p}} \sin(\Omega s)^{\mathbf{q_j^1}} \binom{\mathbf{k}}{\mathbf{p}} ds$$

$$h_{j,\mathbf{p},\mathbf{q}}^2 = \int_0^{2\pi} v_{j,\mathbf{k}}^J \cos(\Omega s)^{\mathbf{p_j^1}} \sin(\Omega s)^{\mathbf{q}} \binom{\mathbf{k}}{\mathbf{p}} ds$$

where $v_{j,\mathbf{k}}^J = a \cos(\Omega s)^{\mathbf{p}} \sin(\Omega s)^{\mathbf{q_j^1}} + b \cos(\Omega s)^{\mathbf{p_j^1}} \sin(\Omega s)^{\mathbf{q}}$. Since the function $\delta(s) = \cos(\Omega s)^{\mathbf{p}+\mathbf{p_j^1}} \sin(\Omega s)^{\mathbf{q}+\mathbf{q_j^1}}$ satisfies $\delta(\pi - s) = -\delta(\pi + s)$, this function integrates to zero, and hence the system above reduces to

$$h_{j,\mathbf{p},\mathbf{q}}^1 = a \frac{1}{\Omega_j} C \binom{\mathbf{k}}{\mathbf{p}}$$

$$h_{j,\mathbf{p},\mathbf{q}}^2 = b C \binom{\mathbf{k}}{\mathbf{p}}$$

for some non-zero constant

$$C = \int_0^{2\pi} \cos(\Omega s)^{2\mathbf{p}} \sin(\Omega s)^{2\mathbf{q_j^1}} ds = \int_0^{2\pi} \cos(\Omega s)^{2\mathbf{p_j^1}} \sin(\Omega s)^{2\mathbf{q}} ds$$

Note that the integral is non-zero since the function inside is positive as all the powers are even.

If $k_j > 0$, then substituting the forms of $x^0(s), v^0(s)$ from (28) in the LHS of (37), and expanding using the binomial theorem, we get that

$$h^1_{j,\mathbf{p},\mathbf{q}} = \frac{1}{\Omega^{\mathbf{q}^1_j}} \int_0^{2\pi} -v^J_{j,\mathbf{k}} \cos(\Omega s)^{\mathbf{p}} \sin(\Omega s)^{\mathbf{q}^1_j} \binom{\mathbf{k}}{\mathbf{p}} ds$$

$$+ \Omega^{\mathbf{p}_j^{-1}} \int_0^{2\pi} v^F_{j,\mathbf{k}_j^{-1}} (-\mathbf{1})^{\mathbf{p}_j^{-1}} \sin(\Omega s)^{\mathbf{p}_j^{-1}} \cos(\Omega s)^{\mathbf{q}^2_j} \binom{\mathbf{k}_j^{-1}}{\mathbf{p}_j^{-1}} ds$$

$$+ \Omega^{\mathbf{p}_j^{-1}} \int_0^{2\pi} v^F_{j,\mathbf{k}^{-1}} (-\mathbf{1})^{\mathbf{p}} \sin(\Omega s)^{\mathbf{p}^1_j} \cos(\Omega s)^{\mathbf{q}} \binom{\mathbf{k}_j^{-1}}{\mathbf{p}} ds$$

$$+ \frac{1}{\Omega^{\mathbf{q}}} \int_0^{2\pi} \left( v^G_{j,\mathbf{k}^{-1}} \cos(\Omega s)^{\mathbf{p}_j^{-1}} \sin(\Omega s)^{\mathbf{q}^2_j} \binom{\mathbf{k}_j^{-1}}{\mathbf{p}_j^{-1}} - v^G_{j,\mathbf{k}^{-1}} \cos(\Omega s)^{\mathbf{p}^1_j} \sin(\Omega s)^{\mathbf{q}} \binom{\mathbf{k}_j^{-1}}{\mathbf{p}} \right) ds$$

$$h^2_{j,\mathbf{p},\mathbf{q}} = \frac{1}{\Omega^{\mathbf{q}}} \int_0^{2\pi} v^J_{j,\mathbf{k}} \cos(\Omega s)^{\mathbf{p}^1_j} \sin(\Omega s)^{\mathbf{q}} \binom{\mathbf{k}}{\mathbf{p}} ds$$

$$+ \Omega^{\mathbf{p}} \int_0^{2\pi} v^F_{j,\mathbf{k}^{-1}} (-\mathbf{1})^{\mathbf{p}_j^{-1}} \sin(\Omega s)^{\mathbf{p}} \cos(\Omega s)^{\mathbf{q}^1_j} \binom{\mathbf{k}_j^{-1}}{\mathbf{p}_j^{-1}} ds$$

$$+ \Omega^{\mathbf{p}} \int_0^{2\pi} v^F_{j,\mathbf{k}^{-1}} (-\mathbf{1})^{\mathbf{p}} \sin(\Omega s)^{\mathbf{p}^2_j} \cos(\Omega s)^{\mathbf{q}_j^{-1}} \binom{\mathbf{k}_j^{-1}}{\mathbf{p}} ds$$

$$+ \frac{1}{\Omega^{\mathbf{q}_j^{-1}}} \int_0^{2\pi} \left( -v^G_{j,\mathbf{k}^{-1}} \cos(\Omega s)^{\mathbf{p}} \sin(\Omega s)^{\mathbf{q}^1_j} \binom{\mathbf{k}_j^{-1}}{\mathbf{p}_j^{-1}} + v^G_{j,\mathbf{k}^{-1}} \cos(\Omega s)^{\mathbf{p}^2_j} \sin(\Omega s)^{\mathbf{q}_j^{-1}} \binom{\mathbf{k}_j^{-1}}{\mathbf{p}} \right) ds$$

Let $g_{\mathbf{k},\mathbf{p}}(s) = \cos(\Omega s)^{\mathbf{p}} \sin(\Omega s)^{\mathbf{k}-\mathbf{p}}$ for all $\mathbf{p} \leq \mathbf{k}$. Crucially, let us assume that $v^J_{j,\mathbf{k}}, v^F_{j,\mathbf{k}}, v^G_{j,\mathbf{k}}$ are all of the form

$$\begin{cases} v^F_{j,\mathbf{k}} = \sum_{\mathbf{r} \leq \mathbf{k}^2_j} \alpha_{\mathbf{k}^2_j,\mathbf{r}} g_{\mathbf{k}^2_j,\mathbf{r}}(s) \\ v^G_{j,\mathbf{k}} = \sum_{\mathbf{r} \leq \mathbf{k}^2_j} \beta_{\mathbf{k}^2_j,\mathbf{r}} g_{\mathbf{k}^2_j,\mathbf{r}}(s) \\ v^J_{j,\mathbf{k}} = \sum_{\mathbf{r} \leq \mathbf{k}^1_j} \gamma_{\mathbf{k}^1_j,\mathbf{r}} g_{\mathbf{k}^1_j,\mathbf{r}}(s) \end{cases} \tag{38}$$

Substituting,

$$h^1_{j,\mathbf{p},\mathbf{q}} = \frac{1}{\Omega^{\mathbf{q}^1_j}} \int_0^{2\pi} - \sum_{\mathbf{r} \leq \mathbf{k}^1_j} \gamma_{\mathbf{k}^1_j,\mathbf{r}} g_{\mathbf{k}^1_j,\mathbf{r}}(s) g_{\mathbf{k}^1_j,\mathbf{p}}(s) \binom{\mathbf{k}}{\mathbf{p}} ds$$

$$+ \Omega^{\mathbf{p}_j^{-1}} \int_0^{2\pi} \left( (-1)^{\mathbf{p}_j^{-1}} \sum_{\mathbf{r} \leq \mathbf{k}^1_j} \alpha_{\mathbf{k}^1_j,\mathbf{r}} g_{\mathbf{k}^1_j,\mathbf{r}}(s) g_{\mathbf{k}^1_j,\mathbf{q}^2_j}(s) \binom{\mathbf{k}_j^{-1}}{\mathbf{p}_j^{-1}} + (-1)^{\mathbf{p}} \sum_{\mathbf{r} \leq \mathbf{k}^1_j} \alpha_{\mathbf{k}^1_j,\mathbf{r}} g_{\mathbf{k}^1_j,\mathbf{r}}(s) g_{\mathbf{k}^1_j,\mathbf{q}}(s) \binom{\mathbf{k}_j^{-1}}{\mathbf{p}} \right) ds$$

$$+ \frac{1}{\Omega^{\mathbf{q}}} \int_0^{2\pi} \left( \sum_{\mathbf{r} \leq \mathbf{k}^1_j} \beta_{\mathbf{k}^1_j,\mathbf{r}} g_{\mathbf{k}^1_j,\mathbf{r}}(s) g_{\mathbf{k}^1_j,\mathbf{p}_j^{-1}}(s) \binom{\mathbf{k}_j^{-1}}{\mathbf{p}_j^{-1}} - \sum_{\mathbf{r} \leq \mathbf{k}^1_j} \beta_{\mathbf{k}^1_j,\mathbf{r}} g_{\mathbf{k}^1_j,\mathbf{r}}(s) g_{\mathbf{k}^1_j,\mathbf{p}^1_j}(s) \binom{\mathbf{k}_j^{-1}}{\mathbf{p}} \right) ds$$

$$h^2_{j,\mathbf{p},\mathbf{q}} = \frac{1}{\Omega^{\mathbf{q}}} \int_0^{2\pi} \sum_{\mathbf{r} \leq \mathbf{k}^1_j} \gamma_{\mathbf{k}^1_j,\mathbf{r}} g_{\mathbf{k}^1_j,\mathbf{r}}(s) g_{\mathbf{k}^1_j,\mathbf{p}^1_j}(s) \binom{\mathbf{k}}{\mathbf{p}} ds$$

$$+ \Omega^{\mathbf{p}} \int_0^{2\pi} \left( (-1)^{\mathbf{p}_j^{-1}} \sum_{\mathbf{r} \leq \mathbf{k}^1_j} \alpha_{\mathbf{k}^1_j,\mathbf{r}} g_{\mathbf{k}^1_j,\mathbf{r}}(s) g_{\mathbf{k}^1_j,\mathbf{q}^1_j}(s) \binom{\mathbf{k}_j^{-1}}{\mathbf{p}_j^{-1}} + (-1)^{\mathbf{p}} \sum_{\mathbf{r} \leq \mathbf{k}^1_j} \alpha_{\mathbf{k}^1_j,\mathbf{r}} g_{\mathbf{k}^1_j,\mathbf{r}}(s) g_{\mathbf{k}^1_j,\mathbf{q}_j^{-1}}(s) \binom{\mathbf{k}_j^{-1}}{\mathbf{p}} \right) ds$$

$$+ \frac{1}{\Omega^{\mathbf{q}_j^{-1}}} \int_0^{2\pi} \left( - \sum_{\mathbf{r} \leq \mathbf{k}^1_j} \beta_{\mathbf{k}^1_j,\mathbf{r}} g_{\mathbf{k}^1_j,\mathbf{r}}(s) g_{\mathbf{k}^1_j,\mathbf{p}}(s) \binom{\mathbf{k}_j^{-1}}{\mathbf{p}_j^{-1}} + \sum_{\mathbf{r} \leq \mathbf{k}^1_j} \beta_{\mathbf{k}^1_j,\mathbf{r}} g_{\mathbf{k}^1_j,\mathbf{r}}(s) g_{\mathbf{k}^1_j,\mathbf{p}^2_j}(s) \binom{\mathbf{k}_j^{-1}}{\mathbf{p}} \right) ds$$

Now, let $\langle f, g \rangle = \int_0^{2\pi} f(s)g(s)ds$ denote the $\ell_2$ inner product. Then, we can rewrite the above system as

$$h_{j,\mathbf{p},\mathbf{q}}^1 = -\frac{1}{\Omega^{\mathbf{q}_{\mathbf{j}}^{\mathbf{1}}}} \sum_{\mathbf{r} \leq \mathbf{k}_{\mathbf{j}}^{\mathbf{1}}} \gamma_{\mathbf{k}_{\mathbf{j}}^{\mathbf{1}},\mathbf{r}} \langle g_{\mathbf{k}_{\mathbf{j}}^{\mathbf{1}},\mathbf{r}}(s), g_{\mathbf{k}_{\mathbf{j}}^{\mathbf{1}},\mathbf{p}}(s) \rangle \binom{\mathbf{k}}{\mathbf{p}}$$

$$+ \Omega^{\mathbf{p}_{\mathbf{j}}^{-1}} \left[ (-\mathbf{1})^{\mathbf{p}_{\mathbf{j}}^{-1}} \sum_{\mathbf{r} \leq \mathbf{k}_{\mathbf{j}}^{\mathbf{1}}} \alpha_{\mathbf{k}_{\mathbf{j}}^{\mathbf{1}},\mathbf{r}} \langle g_{\mathbf{k}_{\mathbf{j}}^{\mathbf{1}},\mathbf{r}}(s), g_{\mathbf{k}_{\mathbf{j}}^{\mathbf{1}},\mathbf{q}_{\mathbf{j}}^{\mathbf{2}}}(s) \rangle \binom{\mathbf{k}_{\mathbf{j}}^{-1}}{\mathbf{p}_{\mathbf{j}}^{-1}} + (-\mathbf{1})^{\mathbf{p}} \sum_{\mathbf{r} \leq \mathbf{k}_{\mathbf{j}}^{\mathbf{1}}} \alpha_{\mathbf{k}_{\mathbf{j}}^{\mathbf{1}},\mathbf{r}} \langle g_{\mathbf{k}_{\mathbf{j}}^{\mathbf{1}},\mathbf{r}}(s), g_{\mathbf{k}_{\mathbf{j}},\mathbf{q}}(s) \rangle \binom{\mathbf{k}_{\mathbf{j}}^{-1}}{\mathbf{p}} \right]$$

$$+ \frac{1}{\Omega^{\mathbf{q}}} \left[ \sum_{\mathbf{r} \leq \mathbf{k}_{\mathbf{j}}^{\mathbf{1}}} \beta_{\mathbf{k}_{\mathbf{j}}^{\mathbf{1}},\mathbf{r}} \langle g_{\mathbf{k}_{\mathbf{j}}^{\mathbf{1}},\mathbf{r}}(s), g_{\mathbf{k}_{\mathbf{j}}^{\mathbf{1}},\mathbf{p}_{\mathbf{j}}^{-1}}(s) \rangle \binom{\mathbf{k}_{\mathbf{j}}^{-1}}{\mathbf{p}_{\mathbf{j}}^{-1}} - \sum_{\mathbf{r} \leq \mathbf{k}_{\mathbf{j}}^{\mathbf{1}}} \beta_{\mathbf{k}_{\mathbf{j}}^{\mathbf{1}},\mathbf{r}} \langle g_{\mathbf{k}_{\mathbf{j}}^{\mathbf{1}},\mathbf{r}}(s), g_{\mathbf{k}_{\mathbf{j}},\mathbf{p}_{\mathbf{j}}^{\mathbf{1}}}(s) \rangle \binom{\mathbf{k}_{\mathbf{j}}^{-1}}{\mathbf{p}} \right]$$

$$h_{j,\mathbf{p},\mathbf{q}}^2 = \frac{1}{\Omega^{\mathbf{q}}} \sum_{\mathbf{r} \leq \mathbf{k}_{\mathbf{j}}^{\mathbf{1}}} \gamma_{\mathbf{k}_{\mathbf{j}}^{\mathbf{1}},\mathbf{r}} \langle g_{\mathbf{k}_{\mathbf{j}}^{\mathbf{1}},\mathbf{r}}(s), g_{\mathbf{k}_{\mathbf{j}}^{\mathbf{1}},\mathbf{p}_{\mathbf{j}}^{\mathbf{1}}}(s) \rangle \binom{\mathbf{k}}{\mathbf{p}}$$

$$+ \Omega^{\mathbf{p}} \left[ (-\mathbf{1})^{\mathbf{p}_{\mathbf{j}}^{-1}} \sum_{\mathbf{r} \leq \mathbf{k}_{\mathbf{j}}^{\mathbf{1}}} \alpha_{\mathbf{k}_{\mathbf{j}}^{\mathbf{1}},\mathbf{r}} \langle g_{\mathbf{k}_{\mathbf{j}}^{\mathbf{1}},\mathbf{r}}(s), g_{\mathbf{k}_{\mathbf{j}},\mathbf{q}_{\mathbf{j}}^{\mathbf{1}}}(s) \rangle \binom{\mathbf{k}_{\mathbf{j}}^{-1}}{\mathbf{p}_{\mathbf{j}}^{-1}} + (-\mathbf{1})^{\mathbf{p}} \sum_{\mathbf{r} \leq \mathbf{k}_{\mathbf{j}}^{\mathbf{1}}} \alpha_{\mathbf{k}_{\mathbf{j}}^{\mathbf{1}},\mathbf{r}} \langle g_{\mathbf{k}_{\mathbf{j}}^{\mathbf{1}},\mathbf{r}}(s), g_{\mathbf{k}_{\mathbf{j}},\mathbf{q}_{\mathbf{j}}^{-1}}(s) \rangle \binom{\mathbf{k}_{\mathbf{j}}^{-1}}{\mathbf{p}} \right]$$

$$+ \frac{1}{\Omega^{\mathbf{q}_{\mathbf{j}}^{-1}}} \left[ - \sum_{\mathbf{r} \leq \mathbf{k}_{\mathbf{j}}^{\mathbf{1}}} \beta_{\mathbf{k}_{\mathbf{j}}^{\mathbf{1}},\mathbf{r}} \langle g_{\mathbf{k}_{\mathbf{j}}^{\mathbf{1}},\mathbf{r}}(s), g_{\mathbf{k}_{\mathbf{j}}^{\mathbf{1}},\mathbf{p}}(s) \rangle \binom{\mathbf{k}_{\mathbf{j}}^{-1}}{\mathbf{p}_{\mathbf{j}}^{-1}} + \sum_{\mathbf{r} \leq \mathbf{k}_{\mathbf{j}}^{\mathbf{1}}} \beta_{\mathbf{k}_{\mathbf{j}}^{\mathbf{1}},\mathbf{r}} \langle g_{\mathbf{k}_{\mathbf{j}}^{\mathbf{1}},\mathbf{r}}(s), g_{\mathbf{k}_{\mathbf{j}}^{\mathbf{1}},\mathbf{p}_{\mathbf{j}}^{\mathbf{2}}}(s) \rangle \binom{\mathbf{k}_{\mathbf{j}}^{-1}}{\mathbf{p}} \right]$$

Now, we will add a few redundant constraints in the system. These are added to ensure that the system has a nice matrix form; they are all of the type $0 = 0$. To do this, we allow $\mathbf{p} \geq \mathbf{0}_{\mathbf{j}}^{-\mathbf{1}}$, instead of $\mathbf{p} \geq \mathbf{0}$. Note that if $p_j = -1$, then $q_j = k_j + 1$ since $\mathbf{p} + \mathbf{q} = \mathbf{k}$. Again, we follow the convention that $\binom{n}{i} = 0$ if $i < 0$ or $i > n$, as well as $g_{\mathbf{k},\mathbf{p}} = 0$ if $\mathbf{p}$ is not between $\mathbf{0}$ and $\mathbf{k}$, both inclusive. Also define $h_{\mathbf{p},\mathbf{q}}^1 = h_{\mathbf{p},\mathbf{q}}^2 = 0$ if either $\mathbf{p}$ or $\mathbf{q}$ are not between $\mathbf{0}$ and $\mathbf{k}$. Thus, all the new constraints added are indeed of the type $0 = 0$.

After these modifications, the system obtained has one constraint corresponding to $h_{\mathbf{p},\mathbf{q}}^t$ for each $\mathbf{0} \leq \mathbf{q} \leq \mathbf{k}_{\mathbf{j}}^{\mathbf{1}}$ (or equivalently $\mathbf{0}_{\mathbf{j}}^{-\mathbf{1}} \leq \mathbf{p} \leq \mathbf{k}$), $\mathbf{p} + \mathbf{q} = \mathbf{k}$, $t = 1,2$ with variables $\alpha_{\mathbf{k}_{\mathbf{j}}^{\mathbf{1}},\mathbf{r}}, \beta_{\mathbf{k}_{\mathbf{j}}^{\mathbf{1}},\mathbf{r}}, \gamma_{\mathbf{k}_{\mathbf{j}}^{\mathbf{1}},\mathbf{r}}$ for $\mathbf{0} \leq \mathbf{r} \leq \mathbf{k}_{\mathbf{j}}^{\mathbf{1}}$. Further, let

$$n_{j,\mathbf{k}} = |D_{\mathbf{k}}| \qquad\qquad D_{\mathbf{k}} = \{\mathbf{r} : \mathbf{0} \leq \mathbf{r} \leq \mathbf{k}\}$$

We will write this system in a matrix form, given by a matrix $A_{j,\mathbf{k}}$ of dimension $2n_{j,\mathbf{k}_{\mathbf{j}}^{\mathbf{1}}} \times 3n_{j,\mathbf{k}_{\mathbf{j}}^{\mathbf{1}}}$ such that

$$A_{j,\mathbf{k}} \begin{bmatrix} \alpha \\ \beta \\ \gamma \end{bmatrix} = \begin{bmatrix} h_j^1 \\ h_j^2 \end{bmatrix}$$

Here $\xi = (\xi_{\mathbf{k}_{\mathbf{j}}^{\mathbf{1}},\mathbf{r}})$ is the vector of dimension $n_{j,\mathbf{k}_{\mathbf{j}}^{\mathbf{1}}}$ for $\xi \in \{\alpha, \beta, \gamma\}$. For notational convenience, we will fix $j$ and $\mathbf{k}$ and denote $A = A_{j,\mathbf{k}}$. We will index rows of $A$ by $(\mathbf{p}, t)$ and columns by $(\mathbf{r}, \xi)$ where $\mathbf{r}, \mathbf{p}_{\mathbf{j}}^{\mathbf{1}} \in D_{\mathbf{k}_{\mathbf{j}}^{\mathbf{1}}}$, $t \in \{1,2\}$, $\xi \in \{\alpha, \beta, \gamma\}$. Further, we will denote by $A_{t,\xi}$ the submatrix of $A$ corresponding to the rows $(\mathbf{p}, t)$ and columns $(\mathbf{r}, \xi)$, that is, $A_{t,\xi}(\mathbf{p}, \mathbf{r}) = A((\mathbf{p}, t), (\mathbf{r}, \xi))$. Matrix $A$ has only $2n_{j,\mathbf{k}}$ non-trivial rows, namely the rows which correspond to $\mathbf{p}$ such that $\mathbf{p} \geq 0$. Hence to show that the system above has a solution, it suffices to prove that matrix $A$ has rank $2n_{j,\mathbf{k}}$.

Define $X, Y$ to be $n_{j,\mathbf{k}} \times n_{j,\mathbf{k}}$ matrices with rows and columns indexed by elements of $D_{\mathbf{k}}$ such that

$$X(\mathbf{p}, \mathbf{r}) = \langle g_{\mathbf{k}_{\mathbf{j}}^{\mathbf{1}},\mathbf{r}}, g_{\mathbf{k}_{\mathbf{j}}^{\mathbf{1}},\mathbf{p}_{\mathbf{j}}^{\mathbf{1}}} \rangle$$

$$Y(\mathbf{p}, \mathbf{r}) = (-\mathbf{1})^{\mathbf{p}_{\mathbf{j}}^{\mathbf{1}}} \langle g_{\mathbf{k}_{\mathbf{j}}^{\mathbf{1}},\mathbf{r}}, g_{\mathbf{k}_{\mathbf{j}}^{\mathbf{1}},\mathbf{k}_{\mathbf{j}}^{\mathbf{1}} - \mathbf{p}_{\mathbf{j}}^{\mathbf{1}}} \rangle$$

Now, assign $\Omega_1 = 1$, $\Omega_j = \frac{M^j - 1}{M - 1}$ for $j > 1$. For this choice of $\Omega_j$'s, it is shown in Turaev [2002] that the functions $g_{\mathbf{k},\mathbf{s}}$ for $\mathbf{0} \leq \mathbf{s} \leq \mathbf{k}$ are linearly independent. It follows from this that the matrices $X$ and $Y$ are full rank. Let $P$ be the permutation matrix that takes row $\mathbf{r}$ of this matrix to row $\mathbf{r}_{\mathbf{j}}^{\mathbf{1}}$

unless $r_j = k_j$, in which case it takes row $\mathbf{r}$ to $\mathbf{s}$ where $s_i = r_i$ for all $i \neq j$ and $s_j = -1$. Thus, for any matrix $M$, $PM(\mathbf{p}, \mathbf{r}) = M(\mathbf{p}_\mathbf{j}^{-1}, \mathbf{r})$ when $p_j \neq -1$, and $PM(\mathbf{p}, \mathbf{r}) = M(\mathbf{p}', \mathbf{r})$ where $p_i' = p_i$ for $i \neq j$ and $p_i' = k_j$ if $p_j = -1$. In particular,

$$PX(\mathbf{p}, \mathbf{r}) = X(\mathbf{p}_\mathbf{j}^{-1}, \mathbf{r}) = \langle g_{\mathbf{k}_\mathbf{j}^\mathbf{1}, \mathbf{r}}, g_{\mathbf{k}_\mathbf{j}^\mathbf{1}, \mathbf{p}} \rangle$$

$$PY(\mathbf{p}, \mathbf{r}) = Y(\mathbf{p}_\mathbf{j}^{-1}, \mathbf{r}) = (-1)^\mathbf{P} \langle g_{\mathbf{k}_\mathbf{j}^\mathbf{1}, \mathbf{r}}, g_{\mathbf{k}_\mathbf{j}^\mathbf{1}, \mathbf{k}_\mathbf{j}^\mathbf{1} - \mathbf{p}} \rangle$$

when $\mathbf{p} \geq \mathbf{0}$. Define $n_{j,\mathbf{k}} \times n_{j,\mathbf{k}}$ diagonal matrices $D_1, D_2, D_3$ such that

$$D_1(\mathbf{p}, \mathbf{p}) = \binom{\mathbf{k}_\mathbf{j}^{-1}}{\mathbf{p}} \qquad D_2(\mathbf{p}, \mathbf{p}) = \binom{\mathbf{k}_\mathbf{j}^{-1}}{\mathbf{p}_\mathbf{j}^{-1}} \qquad D_3(\mathbf{p}, \mathbf{p}) = \binom{\mathbf{k}}{\mathbf{p}}$$

for $\mathbf{0}_\mathbf{j}^{-1} \leq \mathbf{p} \leq \mathbf{k}$. Recalling that $\mathbf{q} = \mathbf{k} - \mathbf{p}$, we see that

$$A_{1,\alpha}(\mathbf{p}, \mathbf{r}) = \Omega^{\mathbf{p}_\mathbf{j}^{-1}} \binom{\mathbf{k}_\mathbf{j}^{-1}}{\mathbf{p}_\mathbf{j}^{-1}} (-1)^{\mathbf{p}_\mathbf{j}^{-1}} \langle g_{\mathbf{k}_\mathbf{j}^\mathbf{1}, \mathbf{r}}, g_{\mathbf{k}_\mathbf{j}^\mathbf{1}, \mathbf{k}_\mathbf{j}^\mathbf{1} - \mathbf{p}_\mathbf{j}^{-1}} \rangle + \Omega^{\mathbf{p}_\mathbf{j}^{-1}} \binom{\mathbf{k}_\mathbf{j}^{-1}}{\mathbf{p}} (-1)^\mathbf{P} \langle g_{\mathbf{k}_\mathbf{j}^\mathbf{1}, \mathbf{r}}, g_{\mathbf{k}_\mathbf{j}^\mathbf{1}, \mathbf{k}_\mathbf{j}^\mathbf{1} - \mathbf{p}_\mathbf{j}^\mathbf{1}} \rangle$$

$$= \Omega^{\mathbf{p}_\mathbf{j}^{-1}} D_2(\mathbf{p}, \mathbf{p}) P^2 Y(\mathbf{p}, \mathbf{r}) - \Omega^{\mathbf{p}_\mathbf{j}^{-1}} D_1(\mathbf{p}, \mathbf{p}) Y(\mathbf{p}, \mathbf{r})$$

$$\Rightarrow A_{1,\alpha} = \Omega^{\mathbf{p}_\mathbf{j}^{-1}} (D_2 P^2 - D_1) Y$$

$$A_{1,\beta}(\mathbf{p}, \mathbf{r}) = \frac{1}{\Omega^\mathbf{q}} \binom{\mathbf{k}_\mathbf{j}^{-1}}{\mathbf{p}_\mathbf{j}^{-1}} \langle g_{\mathbf{k}_\mathbf{j}^\mathbf{1}, \mathbf{r}}, g_{\mathbf{k}_\mathbf{j}^\mathbf{1}, \mathbf{p}_\mathbf{j}^{-1}} \rangle - \frac{1}{\Omega^\mathbf{q}} \binom{\mathbf{k}_\mathbf{j}^{-1}}{\mathbf{p}} \langle g_{\mathbf{k}_\mathbf{j}^\mathbf{1}, \mathbf{r}}, g_{\mathbf{k}_\mathbf{j}^\mathbf{1}, \mathbf{p}_\mathbf{j}^\mathbf{1}} \rangle$$

$$= \frac{1}{\Omega^\mathbf{q}} D_2(\mathbf{p}, \mathbf{p}) P^2 X(\mathbf{p}, \mathbf{r}) - \frac{1}{\Omega^\mathbf{q}} D_1(\mathbf{p}, \mathbf{p}) X(\mathbf{p}, \mathbf{r})$$

$$\Rightarrow A_{1,\beta} = \frac{1}{\Omega^\mathbf{q}} (D_2 P^2 - D_1) X$$

$$A_{1,\gamma}(\mathbf{p}, \mathbf{r}) = -\frac{1}{\Omega^{\mathbf{q}_\mathbf{j}^\mathbf{1}}} \binom{\mathbf{k}}{\mathbf{p}} \langle g_{\mathbf{k}_\mathbf{j}^\mathbf{1}, \mathbf{r}}, g_{\mathbf{k}_\mathbf{j}^\mathbf{1}, \mathbf{p}} \rangle$$

$$= -\frac{1}{\Omega^{\mathbf{q}_\mathbf{j}^\mathbf{1}}} D_3(\mathbf{p}, \mathbf{p}) P X(\mathbf{p}, \mathbf{r})$$

$$\Rightarrow A_{1,\gamma} = -\frac{1}{\Omega^{\mathbf{q}_\mathbf{j}^\mathbf{1}}} D_3 P X$$

$$A_{2,\alpha}(\mathbf{p}, \mathbf{r}) = \Omega^\mathbf{p} \binom{\mathbf{k}_\mathbf{j}^{-1}}{\mathbf{p}_\mathbf{j}^{-1}} (-1)^{\mathbf{p}_\mathbf{j}^{-1}} \langle g_{\mathbf{k}_\mathbf{j}^\mathbf{1}, \mathbf{r}}, g_{\mathbf{k}_\mathbf{j}^\mathbf{1}, \mathbf{k}_\mathbf{j}^\mathbf{1} - \mathbf{p}} \rangle + \Omega^\mathbf{p} \binom{\mathbf{k}_\mathbf{j}^{-1}}{\mathbf{p}} (-1)^\mathbf{P} \langle g_{\mathbf{k}_\mathbf{j}^\mathbf{1}, \mathbf{r}}, g_{\mathbf{k}_\mathbf{j}^\mathbf{1}, \mathbf{k}_\mathbf{j}^\mathbf{1} - \mathbf{p}_\mathbf{j}^\mathbf{2}} \rangle$$

$$= -\Omega^\mathbf{p} D_2(\mathbf{p}, \mathbf{p}) P Y(\mathbf{p}, \mathbf{r}) + \Omega^\mathbf{p} D_1(\mathbf{p}, \mathbf{p}) P^{-1} Y(\mathbf{p}, \mathbf{r})$$

$$\Rightarrow A_{2,\alpha} = \Omega^\mathbf{p} (-D_2 P + D_1 P^{-1}) Y$$

$$A_{2,\beta}(\mathbf{p}, \mathbf{r}) = -\frac{1}{\Omega^{\mathbf{q}_\mathbf{j}^{-1}}} \binom{\mathbf{k}_\mathbf{j}^{-1}}{\mathbf{p}_\mathbf{j}^{-1}} \langle g_{\mathbf{k}_\mathbf{j}^\mathbf{1}, \mathbf{r}}, g_{\mathbf{k}_\mathbf{j}^\mathbf{1}, \mathbf{p}} \rangle + \frac{1}{\Omega^{\mathbf{q}_\mathbf{j}^{-1}}} \binom{\mathbf{k}_\mathbf{j}^{-1}}{\mathbf{p}} \langle g_{\mathbf{k}_\mathbf{j}^\mathbf{1}, \mathbf{r}}, g_{\mathbf{k}_\mathbf{j}^\mathbf{1}, \mathbf{p}_\mathbf{j}^\mathbf{2}} \rangle$$

$$= -\frac{1}{\Omega^{\mathbf{q}_\mathbf{j}^{-1}}} D_2(\mathbf{p}, \mathbf{p}) P X(\mathbf{p}, \mathbf{r}) + \frac{1}{\Omega^{\mathbf{q}_\mathbf{j}^{-1}}} D_1(\mathbf{p}, \mathbf{p}) P^{-1} X(\mathbf{p}, \mathbf{r})$$

$$\Rightarrow A_{2,\beta} = \frac{1}{\Omega^{\mathbf{q}_\mathbf{j}^{-1}}} (-D_2 P + D_1 P^{-1}) X$$

$$A_{2,\gamma}(\mathbf{p}, \mathbf{r}) = \frac{1}{\Omega^\mathbf{q}} \binom{\mathbf{k}}{\mathbf{p}} \langle g_{\mathbf{k}_\mathbf{j}^\mathbf{1}, \mathbf{r}}, g_{\mathbf{k}_\mathbf{j}^\mathbf{1}, \mathbf{p}_\mathbf{j}^\mathbf{1}} \rangle$$

$$= \frac{1}{\Omega^\mathbf{q}} D_3(\mathbf{p}, \mathbf{p}) X(\mathbf{p}, \mathbf{r})$$

$$\Rightarrow A_{2,\gamma} = \frac{1}{\Omega^\mathbf{q}} D_3 X$$

For the above equations to go through as is, we need to check the case when $p_j = -1$, since definitions of $PX$ and $PY$ are different for this case. But, in this case, $D_1(\mathbf{p}, \mathbf{p}) = D_2(\mathbf{p}, \mathbf{p}) = 0$,

and hence the equations hold. Similarly, we need to check the case $p_j = 0$ for blocks $A_{1,\alpha}$ and $A_{1,\beta}$, but again, $D_2(\mathbf{p}, \mathbf{p}) = 0$ and hence the equations hold. Thus, we can write $A$ as

$$\begin{bmatrix} I & 0 \\ 0 & \Omega_j I \end{bmatrix} \begin{bmatrix} D_2 P^2 - D_1 & D_2 P^2 - D_1 & -D_3 P \\ -D_2 P + D_1 P^{-1} & -D_2 P + D_1 P^{-1} & D_3 \end{bmatrix} \begin{bmatrix} \Omega^{\mathbf{p}_j^{-1}} I & 0 & 0 \\ 0 & \frac{1}{\Omega^{\mathbf{q}}} I & 0 \\ 0 & 0 & \frac{1}{\Omega^{\mathbf{q}_j^{1}}} I \end{bmatrix} \begin{bmatrix} Y & 0 & 0 \\ 0 & X & 0 \\ 0 & 0 & X \end{bmatrix}$$

To show that $A$ has rank $2n_{j,\mathbf{k}}$, it suffices to show that the matrix

$$B = \begin{bmatrix} D_2 P^2 - D_1 & -D_3 P \\ -D_2 P + D_1 P^{-1} & D_3 \end{bmatrix}$$

has rank $2n_{j,\mathbf{k}}$. Let us index rows of $B$ using $(\mathbf{p}, s)$ and columns using $(\mathbf{p}, t)$ for $s, t \in \{1, 2\}$. Since $P$ is a permutation matrix, post multiplying by $P$ takes column $\mathbf{r}$ of this matrix to column $\mathbf{r}_j^{-1}$, where the indices cycle whenever they are out of bounds. More specifically,

$$MP(\mathbf{p}, \mathbf{r}) = P^{-1} M^\intercal(\mathbf{r}, \mathbf{p}) = M^\intercal(\mathbf{r}_j^1, \mathbf{p}) = M(\mathbf{p}, \mathbf{r}_j^1).$$

Hence, for a fixed row $(\mathbf{p}, 1)$ the non-zero entries in $B$ are in columns $(\mathbf{p}_j^{-2}, 1), (\mathbf{p}, 1), (\mathbf{p}_j^{-1}, 2)$. Similarly, non-zero entries in the row $(\mathbf{p}, 2)$ are in columns $(\mathbf{p}_j^{-1}, 1), (\mathbf{p}_j^1, 1), (\mathbf{p}, 2)$. Observe that rows $(\mathbf{p}_j^1, 1)$ and $(\mathbf{p}, 2)$ have non-zero entries in the same columns. This gives us a procedure to convert this matrix into a lower triangular matrix using row operations, where indices are ordered using any order $<_R$ that respects

1. $(\mathbf{p}, t) <_R (\mathbf{q}, t)$ if $p_j < q_j$
2. $(\mathbf{p}, 1) <_R (\mathbf{q}, 2)$ for all $\mathbf{0}_j^{-1} \le \mathbf{p}, \mathbf{q} \le \mathbf{k}$

In particular, any lexicographical ordering with highest priority to the $j^{th}$ coordinate works.

Note that only upper triangular non-zero entries using any such ordering are of the type $((\mathbf{p}_j^1, 1), (\mathbf{p}, 2))$. Now, we eliminate these using the following row operations:

$$R(\mathbf{p}_j^1, 1) \leftarrow R(\mathbf{p}_j^1, 1) + C_{\mathbf{p}} R(\mathbf{p}, 2))$$

for all $\mathbf{p}$ such that $0 \le \mathbf{p} \le \mathbf{k}_j^{-1}$. Here

$$C_{\mathbf{p}} = -\frac{B((\mathbf{p}_j^1, 1), (\mathbf{p}, 2))}{B((\mathbf{p}, 2), (\mathbf{p}, 2))} = -\frac{-\binom{\mathbf{k}}{\mathbf{p}_j^1}}{\binom{\mathbf{k}}{\mathbf{p}}} = \frac{\binom{k_j}{p_j+1}}{\binom{k_j}{p_j}} = \frac{k_j - p_j}{p_j + 1}$$

Note that after this set of operations, $B((\mathbf{p}_j^1, 1), (\mathbf{p}, 2)) \leftarrow 0$. On the other hand,

$$B((\mathbf{p}_j^1, 1), (\mathbf{p}_j^1, 1)) \leftarrow B((\mathbf{p}_j^1, 1), (\mathbf{p}_j^1, 1)) + \frac{k_j - p_j}{p_j + 1} B((\mathbf{p}, 2), (\mathbf{p}_j^1, 1))$$

$$= -\binom{\mathbf{k}_j^{-1}}{\mathbf{p}_j^1} + \frac{k_j - p_j}{p_j + 1} \binom{\mathbf{k}_j^{-1}}{\mathbf{p}}$$

$$= \binom{\mathbf{k}_j^{-1}}{\mathbf{p}} \left( -\frac{k_j - p_j - 1}{p_j + 1} + \frac{k_j - p_j}{p_j + 1} \right)$$

$$= \frac{1}{p_j + 1} \binom{\mathbf{k}_j^{-1}}{\mathbf{p}} \ne 0$$

The only non-zero entries in the upper triangle after this operation corresponds to positions $((\mathbf{p}_j^1, 1), (\mathbf{p}, 2))$, for $\mathbf{0}_j^{-1} \le \mathbf{p} \le \mathbf{k}_j^{-1}$, such that $p_j = -1$. To eliminate these, we perform the following row operations:

$$R(\mathbf{p}_j^1, 1) \leftrightarrow R(\mathbf{p}, 2)$$

for all $\mathbf{0}_j^{-1} \le \mathbf{p} \le \mathbf{k}_j^{-1}$ such that $p_j = -1$. Hence,

$$B((\mathbf{p}, 2), (\mathbf{p}, 2)) \leftarrow B((\mathbf{p}_j^1, 1), (\mathbf{p}, 2)) = \binom{\mathbf{k}}{\mathbf{p}_j^1} \ne 0$$

Note that $R(\mathbf{p}, 2) = 0$ since this row corresponds to a dummy constraint. Also, the other two non-zero entries in $R(\mathbf{p_j^1}, 1)$ are in the first half, and hence this does not create any upper triangular entries. Hence, this matrix is in fact lower triangular, in the given ordering $<_R$ of indices.

After the operations, among the diagonal terms, $B((\mathbf{p}, 2), (\mathbf{p}, 2)) \neq 0$ for $\mathbf{0_j^{-1}} \leq \mathbf{p} \leq \mathbf{k}$. Also, $B((\mathbf{p}, 1), (\mathbf{p}, 1)) \neq 0$ for $\mathbf{0_j^1} \leq \mathbf{p} \leq \mathbf{k}$. Therefore, the total number of non-zero diagonal entries is

$$n_{j,\mathbf{k}} \left( \frac{k_j + 1}{k_j} + \frac{k_j - 1}{k_j} \right) = 2n_{j,\mathbf{k}}$$

This proves that the matrix has rank $2n_{j,\mathbf{k}}$, which is the same as the number of non-trivial rows, and hence the system has a solution for any $r_1, r_2$. Consequently, we can always find polynomial functions $J, F, G$ as required. $\qquad\square$

## C   Proof of Lemma 5

*Proof.* From Lemma 2, it suffices to focus on $H$ being a polynomial. We break the time from $\phi$ to 0 for which we want to flow the ODE given by (14) into $(n+1)$ small chunks of length $\tau$, i.e., let $\tau = \phi/(n+1)$. Further, let $A_i = T_{(n-i+1)\tau, (n-i)\tau}$. Then, the time-$\phi$ flow map can be write as the composition of $n+1$ maps, that is

$$T_{\phi, 0} = T_{\tau, 0} \circ \cdots \circ T_{\phi, \phi - \tau} = A_n \circ \cdots \circ A_0$$

Let $\mathcal{C}_0 = T_{0, \phi}(\mathcal{C})$. Let $\mathcal{C}_1, \ldots, \mathcal{C}_{n+1}$ be a sequence of compact sets such that $A_i(\mathcal{C}_i)$ is in the interior of $\mathcal{C}_{i+1}$; by choosing them small enough, we can make $\mathcal{C}_{n+1}$ an arbitrary compact set containing $\mathcal{C}$ in its interior. Below, we treat $A_0, \ldots, A_n$ (and their approximations) as maps $\mathcal{C}_0 \to \mathcal{C}_1 \to \cdots \to \mathcal{C}_{n+1}$, and when we take the $C^1$ norm, we do it on the appropriate compact set. For small enough $\eta$, the $\eta$-discretized maps will stay inside the $\mathcal{C}_i$.

Let $S_i$ denote the time-$2\pi$ flow map obtained by running the ODE system (12) from Lemma 3 above which approximates the map $T_{(n-i+1)\tau, (n-i)\tau} = A_i$. Further, let $S_i'$ denote the map obtained by discretizing the ODE system as in (13) with step size $\eta$. Then, we have that for each $i$, as $\eta \to 0$,

$$\begin{aligned}
\|S_i' - A_i\|_{C^1} &\leq \|S_i' - S_i + S_i - A_i\|_{C^1} \\
&\leq \|S_i' - S_i\|_{C^1} + \|S_i - A_i\|_{C^1} \\
&\leq O(\eta) + O(\tau^2) \qquad\qquad\qquad \text{(by Lemmas 3 and 4)}
\end{aligned}$$

We choose $\eta = \tau^2$. Using the definition of $C^1$ norm, this implies that

$$\|S_i' - A_i\| = O(\tau^2) \qquad\qquad \|DS_i' - DA_i\| = O(\tau^2),$$

where $\|\cdot\|$ denotes $L^\infty$ norm on $\mathcal{C}_i$; for matrix-valued functions $M(x)$ on $\mathcal{C}_i$, $\|M\| = \sup_{x \in \mathcal{C}_i} \|M(x)\|_2$, where $\|\cdot\|_2$ denotes spectral norm. Again, using the definition of the $C^1$ norm,

$$\begin{aligned}
&\|A_n \circ \cdots \circ A_0 - S_n' \circ \cdots \circ S_0'\|_{C^1} \\
&\leq \|A_n \circ \cdots \circ A_0 - S_n' \circ \cdots \circ S_0'\| + \|D(A_n \circ \cdots \circ A_0) - D(S_n' \circ \cdots \circ S_0')\|
\end{aligned}$$

We will bound each term individually. For the first term, note that

$$\begin{aligned}
&\|A_n \circ \cdots \circ A_0 - S_n' \circ \cdots \circ S_0'\| \\
&\leq \|A_n \circ \cdots \circ A_1 \circ A_0 - A_n \circ \cdots \circ A_1 \circ S_0'\| + \|A_n \circ \cdots \circ A_1 \circ S_0' - S_n' \circ \cdots \circ S_1' \circ S_0'\| \\
&\qquad\qquad\qquad\qquad\qquad\qquad\qquad\qquad\qquad\qquad\qquad\qquad \text{(by triangle inequality)} \\
&= \|T_{\phi - \tau, 0} \circ A_0 - T_{\phi - \tau} \circ S_0'\| + \|A_n \circ \cdots \circ A_1 \circ S_0' - S_n' \circ \cdots \circ S_1' \circ S_0'\| \\
&\leq \|DT_{\phi - \tau, 0}\| \|S_0' - A_0\| + \|A_n \circ \cdots \circ A_1 \circ S_0' - S_n' \circ \cdots \circ S_1' \circ S_0'\| \\
&\leq O(\tau^2) + \|A_n \circ \cdots \circ A_1 \circ S_0' - S_n' \circ \cdots \circ S_1' \circ S_0'\| \qquad\qquad\qquad\qquad (39)
\end{aligned}$$

Observe that

$$\begin{aligned}
&\sup_x \|A_n \circ \cdots \circ A_1 \circ S_0'(x) - S_n' \circ \cdots \circ S_1' \circ S_0'(x)\| \\
&= \sup_{y = S_0'(x)} \|A_n \circ \cdots \circ A_1(y) - S_n' \circ \cdots \circ S_1'(y)\| \\
&\leq \sup_y \|A_n \circ \cdots \circ A_1(y) - S_n' \circ \cdots \circ S_1'(y)\| \\
&= \|A_n \circ \cdots \circ A_1(y) - S_n' \circ \cdots \circ S_1'(y)\| \qquad\qquad\qquad\qquad\qquad\qquad (40)
\end{aligned}$$

Using (40), (39), and induction, we get that

$$\|A_n \circ \cdots \circ A_0 - S'_n \circ \cdots \circ S'_0\| \le O(n\tau^2)$$

Now, we bound the derivatives:

$$
\begin{aligned}
&\|D(A_n \circ \cdots \circ A_0) - D(S'_n \circ \cdots \circ S'_0)\| \\
&\le \|D(A_n \circ \cdots \circ A_1 \circ A_0) - D(A_n \circ \cdots \circ A_1 \circ S'_0)\| \\
&\quad + \|D(A_n \circ \cdots \circ A_1 \circ S'_0) - D(S'_n \circ \cdots \circ S'_1 \circ S'_0)\| \qquad \text{(by triangle inequality)} \\
&= \sup_x \|DT_{\phi-\tau,0}|_{A_0(x)} DA_0(x) - DT_{\phi-\tau,0}|_{S'_0(x)} DS'_0(x)\| \\
&\quad + \sup_x \|D(A_n \circ \cdots \circ A_1)|_{S'_0(x)} DS'_0(x) - D(S'_n \circ \cdots \circ S'_1)|_{S'_0(x)} DS'_0)(x)\| \quad \text{(by chain rule)} \\
&\le \sup_x \|DT_{\phi-\tau,0}|_{A_0(x)} DA_0(x) - DT_{\phi-\tau,0}|_{S'_0(x)} DA_0(x)\| \\
&\quad + \sup_x \|DT_{\phi-\tau,0}|_{S'_0(x)} DA_0(x) - DT_{\phi-\tau,0}|_{S'_0(x)} DS'_0(x)\| \qquad \text{(by triangle inequality)} \\
&\quad + \|DS'_0\| \|D(A_n \circ \cdots \circ A_1) - D(S'_n \circ \cdots \circ S'_1)\| \qquad\qquad\qquad\qquad\qquad (41) \\
&\le \sup_x \|DT_{\phi-\tau,0}|_{A_0(x)} - DT_{\phi-\tau,0}|_{S'_0(x)}\| \|DA_0\| \\
&\quad + \sup_x \|DT_{\phi-\tau,0}|_{S'_0(x)}\| \|DA_0 - DS_0\| \\
&\quad + \|DS'_0\| \|D(A_n \circ \cdots \circ A_1) - D(S'_n \circ \cdots \circ S'_1)\| \\
&\le \|D^2 T_{\phi-\tau,0}\| \|S'_0 - A'_0\| \|DA_0\| + \|DT_{\phi-\tau,0}\| \|DA_0 - DS'_0\| \\
&\quad + \|DS'_0\| \|D(A_n \circ \cdots \circ A_1) - D(S'_n \circ \cdots \circ S'_1)\| \\
&\le O(\tau^2) + \Big(\|DA_0\| + O(\tau^2)\Big) \|D(A_n \circ \cdots \circ A_1) - D(S'_n \circ \cdots \circ S'_1)\| \qquad (42)
\end{aligned}
$$

where, for a 3-tensor $\mathcal{T}$, we define $\|\mathcal{T}\| = \sup_{\|u\|=1} \|\mathcal{T}u\|_2$, where $\|\mathcal{T}u\|_2$ is the spectral norm of the matrix $\mathcal{T}u$, and we define $\|D^2 T_{\phi-\tau,0}\| = \sup_x \|D^2 T_{\phi-\tau,0}(x)\|$. In the last step, we use the fact that $\|DT_{s,t}\|, \|D^2 T_{s,t}\|$ are bounded for all $s, t > 0$; this follows from Lemma 9 below. (Alternatively, note that $\|DT_{s,t}\|$ can also be more directly bounded by Theorem 6.)

In the above, (41) follows using an argument similar to (40), (42) follows since $\|DA_0 - DS'_0\| = O(\tau^2)$. Further, differentiating (46), we get

$$DA_0 = I + \tau D_{(x,v)} F(x, v, t) + O(\tau^2)$$

where $F$ denotes the defining equation of the ODE system in (14). Therefore, we get

$$\|DA_0\| \le 1 + \tau L + O(\tau^2)$$

where $L$ is the upper bound on $\|Df\|$ over all the appropriate compact sets. Using this bound and induction, we get that

$$\|D(A_n \circ \cdots \circ A_0) - D(S'_n \circ \cdots \circ S'_0)\| \le O(n\tau^2)(1 + \tau L + O(\tau^2))^n = O(n\tau^2 e^{n\tau L})$$

for small enough $\tau$. Substituting $n\tau = \phi$, we get the overall $C^1$ bound of

$$\|A_n \circ \cdots \circ A_0 - S'_n \circ \cdots \circ S'_0\|_{C^1} = O(\phi \tau e^{\phi L}).$$

Now, we can choose $\tau$ small enough so that the two maps are $\epsilon_1$-close, finishing the proof.

Concretely, we can write each $S'_i$ as a composition of affine-coupling maps (which constitute the $f_1, \ldots, f_N$ in the lemma statement). In this manner, we can compose these compositions of affine coupling maps over each $\tau$-sized chunk of time so as to get a map which is overall close to the required flow map. □

**Lemma 9.** *Consider the ODE $\frac{d}{dt} x(t) = F(x(t), t)$ for $F(x, t)$ that is $C^\ell$ in $x \in \mathbb{R}^d$ and continuous in $t$. Let $\mathcal{C}$ be a compact set and suppose solutions exist for any $(x(0), v(0)) \in \mathcal{C}$ up to time $T$. Let $T_{s,t}$ be the flow map from time $s$ to time $t$, for any $0 \le s, t \le T$. Then for any $0 \le r \le \ell$, $D^r T_{s,t}$ is bounded on $T_s(\mathcal{C})$.*

*Proof.* Let $\partial_{i_1 \cdots i_r} = \frac{\partial^r}{\partial x_{i_1} \cdots \partial x_{i_r}}$. Using the chain rule as in Lemma 12, we find by induction that

$$\frac{d}{dt} \partial_{i_1 \cdots i_r}(T_t(x)) = \sum_{i=1}^{d} \partial_i F(x(t), t) \partial_{i_1 \cdots i_r}(T_t(x)_i) + G(DF, \dots, D^r F, DT_t, \dots, D^{r-1} T_t). \tag{43}$$

for some polynomial $G$. For $r = 1$, the differential equation is given by Lemma 12. By a Grönwall argument, a bound on $DF$ gives an upper and lower bound on the singular values of $DT_t$ as in (23). We use induction on $r$; for $r > 1$, let $v(t)$ be equal to $(\partial_{i_1 \cdots i_r}(T_t(x)))_{i_1 \cdots i_r}$ written as one large vector. By the chain rule and (43),

$$\frac{d}{dt} \|v(t)\|^2 \leq \langle |v(t)|, A|v(t)| + b \rangle \leq \left( \sigma_{\max}(A) + \frac{1}{2} \right) \|v(t)\|^2 + \frac{1}{2} \|b\|^2$$

for some $A, b$ depending on $DF, \dots, D^r F, DT_t, \dots, D^{r-1} T_t$, where $\sigma_{\max}$ denotes the maximum singular value and $|v|$ denotes entrywise absolute value. Grönwall's inequality (Lemma 15) applied to $\|v(t)\|^2$ then gives bounds on $\|v(t)\|^2$ and hence $\left| \frac{d}{dt} \partial_{i_1 \cdots i_r}(T_t(x)) \right|$. This shows $D^r T_{s,t}$ is bounded when $s \leq t$ (by starting the flow at time $s$).

When $s > t$, note that the computation of the $r$th derivative of an inverse map involves up-to-$r$ derivatives of the forward map, and inverses of the first derivative. As we have a lower bound on the singular value of $DF$, this implies that $D^r T_{s,t}$ is bounded. $\square$

# D    Technical Tools

## D.1    Proof of Lemma 4

We consider a more general ODE than the specific one in (12), of the form

$$\begin{cases} \frac{d}{dt}(x(t)) = f(x(t), v(t), t) \\ \frac{d}{dt}(v(t)) = g(x(t), v(t), t) \end{cases} \tag{44}$$

where $f, g$ are $C^2$ functions in $x, v, t$. Given a compact set $\mathcal{C}$, suppose that the solutions are well-defined for any $(x(0), v(0)) \in \mathcal{C}$ up to time $T$. Consider discretizing these ODEs into steps of size $\eta$, as follows:

$$\begin{cases} \widetilde{T}_i^x(X_i) = X_{i+1} = X_i + \eta f(X_i, V_{i+1}, t_i) \\ \widetilde{T}_i^v(V_i) = V_{i+1} = V_i + \eta g(X_i, V_i, t_i) \end{cases} \tag{45}$$

where $t_i = i\eta$. We call this the alternating Euler update. The actual flow maps are given by

$$\begin{cases} T_i^x(x_i) = x_{i+1} = x_i + \eta f(x_i, v_i, t_i) + \int_{i\eta}^{(i+1)\eta} \int_{i\eta}^t x''(s) \, ds \, dt \\ T_i^v(v_i) = v_{i+1} = v_i + \eta g(x_i, v_i, t_i) + \int_{i\eta}^{(i+1)\eta} \int_{i\eta}^t v''(s) \, ds \, dt \end{cases} \tag{46}$$

We bound the local truncation error. This consists of two parts. First, we have the integral terms in (46):

$$\left\| \begin{bmatrix} \int_{i\eta}^{(i+1)\eta} \int_{i\eta}^t x''(s) \, ds \, dt \\ \int_{i\eta}^{(i+1)\eta} \int_{i\eta}^t v''(s) \, ds \, dt \end{bmatrix} \right\| \leq \frac{1}{2} \eta^2 \max_{s \in [0, t_i]} \left\| \begin{bmatrix} x''(s) \\ v''(s) \end{bmatrix} \right\|. \tag{47}$$

Second we bound the error from using $\widetilde{v}_{i+1} := v_i + \eta g(x_i, v_i, t_i)$ instead of $v_i$ in the $x$ update,

$$\|\eta[f(x_i, v_i + \eta g(x_i, v_i, t_i), t_i) - f(x_i, v_i, t_i)]\| \leq \left\| \eta \int_0^\eta D_v f(x_i, v_i + s g(x_i, v_i, t_i), t_i) g(x_i, v_i, t_i) \, ds \right\|$$

$$\leq \eta^2 \max_{\mathcal{C}'} \|D_v f\| \max_{\mathcal{C}'} \|g\|. \tag{48}$$

where $D_v f(x, v, t)$ denotes the Jacobian in the $v$ variables (rather than the directional derivative), and where we define

$$\mathcal{C}' := \{(x, v + s g(x, v, t), t) : (x, v) = T_t(x_0, v_0) \text{ for some } (x_0, v_0) \in \mathcal{C}, 0 \leq s \leq T\},$$

which ensures that it contains $(x_i, v_i + sg(x_i, v_i, t_i), t_i)$ and $(x_i, v_i, t_i)$. The local truncation error is then at most the sum of (47) and (48).

Supposing that $\begin{bmatrix} f \\ g \end{bmatrix}$ is $L$-Lipschitz in $(x, v) \in \mathbb{R}^{2d}$ for each $t$, we obtain by a standard argument (similar to the proof for the usual Euler's method, see e.g., [Ascher and Greif, 2011, §16.2]) that the global error at any step is bounded by

$$\left\| \begin{bmatrix} \widetilde{x}_i \\ \widetilde{v}_i \end{bmatrix} - \begin{bmatrix} x_i \\ v_i \end{bmatrix} \right\| \leq \eta \cdot \frac{e^{Lt_i} - 1}{L} \left( \max_{\mathcal{C}'} \|D_v f\| \max_{\mathcal{C}'} \|g\| + \frac{1}{2} \max_{s \in [0, t_i]} \left\| \begin{bmatrix} x''(s) \\ v''(s) \end{bmatrix} \right\| \right). \tag{49}$$

In the case when $\begin{bmatrix} f \\ g \end{bmatrix}$ is not globally Lipschitz, we show that we can restrict the argument to a compact set on which it is Lipschitz. Let $\mathcal{C}''$ be a compact set which contains $\{(x, v, t) : (x, v) = T_t(x_0, v_0)$ for some $(x_0, v_0) \in \mathcal{C}, 0 \leq s \leq T\}$ in its interior. Apply the argument to $\hat{f}$ and $\hat{g}$ which are defined to be equal to $f, g$ on $\mathcal{C}''$, and are globally Lipschitz. Then the error bound applies to the system defined by $\hat{f}, \hat{g}$. Hence, for small enough step size, the trajectory of the discretization stays inside $\mathcal{C}''$, and is the same as that for the system defined by $f, g$. Then (49) holds for small enough $\eta$ and $L$ equal to the Lipschitz constant in $(x, v)$ on $\mathcal{C}''$.

To get a bound in $C^1$ topology, we need to bound the derivatives of these maps as well. Let $T_{s,t}(x, v)$ denote the flow map of system (44). Let $h(x, v, t) = (f(x, v, t), g(x, v, t))$. Now, consider the system of ODEs

$$\begin{cases} \frac{d}{dt}(x(t)) = f(x(t), v(t), t) \\ \frac{d}{dt}(v(t)) = g(x(t), v(t), t) \\ \frac{d}{dt}(\alpha(t)) = D_{(x,v)} f(x(t), v(t), t) \begin{bmatrix} \alpha(t) \\ \beta(t) \end{bmatrix} \\ \frac{d}{dt}(\beta(t)) = D_{(x,v)} g(x(t), v(t), t) \begin{bmatrix} \alpha(t) \\ \beta(t) \end{bmatrix} \end{cases} \tag{50}$$

where $\alpha(t), \beta(t)$ are $d \times 2d$ matrices. Note that setting $\begin{bmatrix} \alpha(0) \\ \beta(0) \end{bmatrix} = \mathrm{I}_{2d}$ and $\begin{bmatrix} \alpha(t) \\ \beta(t) \end{bmatrix} = D_{(x,v)} T_{0,t}(x(0), v(0))$ satisfies (50) by Lemma 12.

Now we claim that applying the alternating Euler update to $(x, \alpha), (v, \beta)$, the resulting $(\alpha_i, \beta_i)$ is exactly the Jacobian of the flow map that arises from alternating Euler applied to $x, v$. This means that we can bound the errors for $\alpha, \beta$ using the bound for the alternating Euler method.

The claim follows from noting that the alternating Euler update on $\alpha, \beta$ is

$$\alpha_{i+1} = (\mathrm{I}_d, O) + D_{(x,v)} f(x_i, v_{i+1}, t_i) \begin{bmatrix} \alpha_i \\ \beta_{i+1} \end{bmatrix}$$

$$\beta_{i+1} = (O, \mathrm{I}_d) + D_{(x,v)} f(x_i, v_i, t_i) \begin{bmatrix} \alpha_i \\ \beta_i \end{bmatrix},$$

which is the same recurrence that is obtained from differentiating $X_{i+1}, V_{i+1}$ in (45) with respect to $X_0, V_0$, and using the chain rule.

Thus we can apply (49) to get a bound for the Jacobians of the flow map. The constants in the $O(\eta)$ bound depend on up to the second derivatives of the $x, v, \alpha, \beta$ for the true solution, Lipschitz constants for $\begin{bmatrix} f \\ g \end{bmatrix}, D \begin{bmatrix} f \\ g \end{bmatrix}$ (on a suitable compact set), and bounds for $D_v f, g, D_v D_{(x,v)} f, D_{(x,v)} g$ (on a suitable compact set).

## D.2 Wasserstein bounds

**Lemma 10.** *Given two distributions $p, q$ and a function $g$ with Lipschitz constant $L = \mathrm{Lip}(g)$,*

$$W_1(g_\# p, g_\# q) \leq L W_1(p, q)$$

*Proof.* Let $\epsilon > 0$. Then there exists a coupling $(x, t) \sim \gamma$ such that

$$\int \|x - y\|_2 d\gamma(x, y) \leq W_1(p, q) + \epsilon$$

Consider the coupling $(x', y')$ given by $(x', y') = (g(x), g(y))$ where $(x, y) \sim \gamma$. Then

$$W_1(g_\# p, g_\# q) \leq \int \|g(x) - g(y)\|_2 \, d\gamma(x, y)$$
$$\leq \mathrm{Lip}(g) \int \|x - y\| \, d\gamma(x, y)$$
$$\leq L W_1(p, q) + L\epsilon.$$

Since this holds for all $\epsilon > 0$, we get that

$$W_1(g_\# p, g_\# q) \leq L W_1(p, q)$$

$\square$

**Lemma 11.** *Given two functions $f, g : \mathbb{R}^d \to \mathbb{R}^d$ that are uniformly $\epsilon_1$-close over a compact set $\mathcal{C}$ in $C^1$ topology, and a probability distribution $p$,*

$$W_1(f_\#(p|_\mathcal{C}), g_\#(p|_\mathcal{C})) \leq \epsilon_1$$

*Proof.* Consider the coupling $\gamma$, where a sample $(x, y) \sim \gamma$ is generated as follows: first, we sample $z \sim p|_\mathcal{C}$, and then compute $x = f(z)$, $y = g(z)$. By definition of the pushforward, the marginals of $x$ and $y$ are $f_\#(p|_\mathcal{C})$ and $g_\#(p|_\mathcal{C})$ respectively. However, we are given that for this $\gamma$, $\|x - y\| \leq \epsilon_1$ uniformly. Thus, we can conclude that

$$W_1(f_\#(p|_\mathcal{C}), g_\#(p|_\mathcal{C})) \leq \int_{\mathbb{R}^d \times \mathbb{R}^d} \|x - y\|_2 \, d\gamma(x, y)$$
$$\leq \int_{\mathbb{R}^d \times \mathbb{R}^d} \epsilon_1 \, d\gamma(x, y) = \epsilon_1$$

$\square$

### D.3 Proof of Lemma 6

*Proof.* Fix any $R > 0$, and set $\mathcal{C} = B(0, R)$. Consider the coupling $(X, Y) \sim \gamma$, where a sample $(X, Y)$ is generated as follows: we first sample $X \sim p^* = \mathcal{N}(0, \mathrm{I}_{2d})$. If $X \in B(0, R)$, then we set $Y = X$. Else, we draw $Y$ from $p^*|_\mathcal{C}$. Clearly, the marginal of $\gamma$ on $X$ is $p$. Furthermore, since $p^*$ and $p^*|_\mathcal{C}$ are proportional within $\mathcal{C}$, the marginal of $\gamma$ on $Y$ is $p^*|_\mathcal{C}$. Then, we have that

$$W_1(p^*, p^*|_\mathcal{C}) \leq \int_{\mathbb{R}^{2d} \times \mathcal{C}} \|x - y\| d\gamma$$
$$= \cancel{\int_{\mathcal{C} \times \mathcal{C}} \|x - y\| d\gamma} + \int_{\mathbb{R}^{2d} \backslash \mathcal{C} \times \mathcal{C}} \|x - y\| d\gamma$$
$$= \int_{\mathbb{R}^{2d} \backslash \mathcal{C} \times \mathcal{C}} \|x - y\| d\gamma$$
$$\leq \int_{\mathbb{R}^{2d} \backslash \mathcal{C} \times \mathcal{C}} (\|x\| + \|y\|) d\gamma$$
$$\leq \int_{\mathbb{R}^{2d} \backslash \mathcal{C} \times \mathcal{C}} (\|x\| + R) d\gamma$$
$$\leq \int_{\mathbb{R}^{2d} \backslash \mathcal{C} \times \mathcal{C}} (\|x\| + R) d\gamma$$
$$= \int_{\mathbb{R}^{2d} \backslash \mathcal{C}} (\|x\| + R) dp^*$$
$$\leq \int_{\mathbb{R}^{2d} \backslash \mathcal{C}} 2\|x\| dp^* = \frac{2}{\sqrt{2\pi}} \int_{\mathbb{R}^{2d} \backslash \mathcal{C}} \|x\| e^{-\frac{\|x\|^2}{2}} dx$$

Now, note that $\int_{\mathbb{R}^{2d}} \|x\| e^{-\frac{\|x\|^2}{2}} \, dx < \infty$. Hence, by the Dominated Convergence Theorem,

$$\lim_{R \to \infty} \int_{\mathbb{R}^{2d} \setminus B(0,R)} \|x\| e^{-\frac{\|x\|^2}{2}} \, dx = 0.$$

Thus, given any $\delta > 0$, we can choose $R$ large enough so that the integral above is smaller than $\delta$, which concludes the proof. $\qquad\square$

### D.4 Derivatives of flow maps

We state and prove a technical lemma about the ODE that the derivative of a flow map satisfies.

**Lemma 12.** *Suppose $x_t = x(t)$ satisfies the ODE*

$$\dot{x} = F(x, t)$$

*with flow map $T(x,t) : \mathbb{R}^n \times \mathbb{R} \to \mathbb{R}^n$. Suppose $\alpha(t)$ be the derivative of the map $x \mapsto T(x,t)$ at $x_0$, then $\alpha(t)$ satisfies*

$$\dot{\alpha} = DF(x_t, t)\alpha$$

*with $\alpha(0) = \mathrm{I}$.*

*Proof.* Let $T_t(x) = T(x, t)$. Then $T_t$ satisfies

$$T_t(x_0) = \int_0^t F(x_s, s) \, ds.$$

Differentiating, we get

$$
\begin{aligned}
\alpha(t) = DT_t(x_0) &= \int_0^t D(F(x_s, s)) \, ds \\
&= \int_0^t DF(x_s, s) DT_s(x_0) \, ds \qquad\qquad \text{by chain rule} \\
&= \int_0^t DF(x_s, s)\alpha(s) \, ds.
\end{aligned}
$$

Now, looking at the derivative with respect to $t$, we get

$$\dot{\alpha} = DF(x_t, t)\alpha,$$

which is the required result. $\qquad\square$

### D.5 Solving Perturbed ODEs

In this section, we state a result about finding approximate solutions of perturbed differential equations. Consider the ODE having the following general form:

$$\dot{x} = Ax + \epsilon g(x, t)$$

The reason we are concerned with this ODE is that the ODE given by Equation (12) has precisely this form, namely with $x \equiv \begin{bmatrix} x \\ v \end{bmatrix}$, $A \equiv \begin{bmatrix} 0 & \mathrm{I}_d \\ -\mathrm{diag}(\Omega^2) & 0 \end{bmatrix}$ and $\epsilon g(x,t) \equiv -\tau \begin{bmatrix} F(v,t) \odot x \\ J(x,t) + G(x,t) \odot v \end{bmatrix}$.

Let $T^x : \mathbb{R} \times \mathbb{R}^n \to \mathbb{R}^n$ be the time $t$ flow map for this ODE. We will find a flow map $T^y : \mathbb{R} \times \mathbb{R}^n \to \mathbb{R}^n$ such that the maps $T_t^x$ defined by $T_t^x(x) = T^x(t, x)$ and the map $T_t^y$ defined by $T_t^y(y) = T^y(t, y)$ are uniformly $\epsilon$-close over $\mathcal{C}$ in $C^r$ topology for all $0 \leq t \leq 2\pi$. That is,

$$\sup_x \quad \|T_t^x(x) - T_t^y(x)\| + \|DT_t^x(x) - DT_t^y(x)\| + \cdots + \|D^r T_t^x(x) - D^r T_t^y(x)\|$$

is small, for all $t \in [0, 2\pi]$. Here $D^r$ denotes the $r$-th derivative, and the norms are defined inductively as follows: for a $r$-tensor $\mathcal{T}$, we let $\|\mathcal{T}\| = \sup_{\|u\|=1} \|\mathcal{T}u\|$; here $\mathcal{T}u$ is a $(r-1)$-tensor. (The choice of norm is not important; we choose this for convenience.)

**Lemma 13.** *Consider the ODE*

$$\frac{d}{dt}x(t) = F(x(t), t) + \epsilon G(x(t), t) \tag{51}$$

*where $x : [0, t_{\max}] \to \mathbb{R}^n$, $F, G : \mathbb{R}^n \times \mathbb{R} \to \mathbb{R}^n$, and $F(x, t)$, $G(x, t)$ are $C^1$, and $F$ is $L$-Lipschitz. Let $\mathcal{C}$ be a compact set, and suppose that for all $x_0 \in \mathcal{C}$, solutions to (51) with $x(0) = x_0$ exist for $0 \leq t \leq t_{\max}$ and $\epsilon = 0$. Then there exists $\epsilon_0$ such that solutions to (51) with $x(0) = x_0$ exist for $0 \leq t \leq t_{\max}$ and $0 \leq \epsilon < \epsilon_0$.*

*Moreover, letting $x^{(\epsilon)}(t)$ be the solution with given $\epsilon$, we have that as $\epsilon \to 0$, $\left\| x^{(\epsilon)}(t) - x^{(0)}(t) \right\| = O(\epsilon)$, where the constants in the $O(\cdot)$ depend only on $L$ and $\max_{0 \leq t \leq t_{\max}, x_0 \in \mathcal{C}} \left\| G(x^{(0)}(t), t) \right\|$ (the maximum of $G$ on the $\epsilon = 0$ trajectories).*

*Proof.* Let $T^\epsilon(t, x_0)$ be the flow map of (51). Let $\mathcal{K} = T^0(\mathcal{C} \times [0, t_{\max}])$ be the image of $\mathcal{C} \times [0, t_{\max}]$ under the flow map $T^0$. Since $F$ is $C^1$, $T^0$ is $C^1$, which implies that $\mathcal{K}$ is bounded. Fix some $\epsilon_2 > 0$. Let $B(\mathcal{K}, r)$ denote the set

$$B(\mathcal{K}, r) = \{(x, t) \in \mathbb{R}^n \times [0, t_{\max}] : d(\mathcal{K}, x) \leq r\}$$

Let $\mathcal{K}_2 = B(\mathcal{K}, \epsilon_2)$. Note that since $\mathcal{K}$ is compact, so is $\mathcal{K}_2$. Let

$$M = \max\left\{ \sup_{(x,t) \in \mathcal{K}_2 \times [0, t_{\max}]} \|F(x, t)\|, \sup_{(x,t) \in \mathcal{K}_2 \times [0, t_{\max}]} \|G(x, t)\| \right\}$$

$M$ is finite since $\mathcal{K}_2$ is compact and $F, G$ are $C^1$.

Let $h : \mathbb{R} \to \mathbb{R}$ be a 1-Lipschitz $C^1$ function such that

$$h(x) = x \text{ if } |x| \leq M$$
$$|h(x)| \leq 2M \text{ for all } x.$$

Let $h_n : \mathbb{R}^n \to \mathbb{R}^n$ be defined as $h_n(x) = \frac{x}{\|x\|} h(\|x\|)$. Then $h_n(x)$ is also $C^1$ and is the identity function on $B(0, M)$. Let $F_1 = h_n \circ F$ and let $G_1 = h_n \circ F$. Then $F_1, G_1$ are $C^1$ functions such that $\|F_1\|, \|G_1\| \leq 2M$. Further, $F_1$ is $L$-Lipschitz. Now, we look at the ODE

$$\frac{d}{dt}x(t) = F_1(x(t), t) + \epsilon G_1(x(t), t) \tag{52}$$

Since $F_1, G_1$ are $C^1$, note that the function $H_1(x, \epsilon, t) = F_1(x, t) + \epsilon G_1(x, t)$ is $C^1$ in $x, t, \epsilon$. Therefore, using the existence theorem for parametric ODEs (Theorem 1.2, Chicone [2006]), there is a $\epsilon_1, t_1 > 0$ such that solutions $x_1^{(\epsilon)}(t)$ to (52) exist for all $x_0 \in \mathcal{C}, \epsilon < \epsilon_1$ and $t < t_1$. Further, the extensibility result for the ODEs (Theorem 1.4, Chicone [2006]) states that if $t_1$ is largest such value for which such solutions exist, then there exists a $x_0 \in \mathcal{C}$ and $\epsilon < \epsilon_1$ such that $\lim_{t \to t_1} \|x_1^{(\epsilon)}(t)\| = \infty$.

Now, we will bound $\|x_1^{(\epsilon)} - x_1^{(0)}\|$ for $t < t_1$. Define $\alpha = x_1^{(0)} - x_1^{(\epsilon)}$. Then $\alpha(t)$ satisfies

$$\frac{d}{dt}\alpha(t) = F_1(x_1^{(0)}(t), t) - F_1(x_1^{(\epsilon)}(t), t) - \epsilon G_1(x_1^{(\epsilon)}(t), t)$$

Therefore,

$$\frac{d}{dt}\|\alpha(t)\|^2 \leq 2\|\alpha(t)\| \left\| \frac{d}{dt}\alpha(t) \right\|$$

$$\leq 2\|\alpha(t)\| \|F_1(x_1^{(0)}(t), t) - F_1(x_1^{(\epsilon)}(t), t) - \epsilon G_1(x_1^{(\epsilon)}(t), t)\|$$

$$\leq 2\|\alpha(t)\| (L\|\alpha(t)\| + 2\epsilon M)$$

$$\leq 2L\|\alpha(t)\|^2 + 4\epsilon M \|\alpha(t)\|$$

$$\implies \frac{d}{dt}\|\alpha(t)\| \leq \frac{1}{2} \|\alpha(t)\|^{-1} \frac{d}{dt}\|\alpha(t)\|^2 \leq L\|\alpha(t)\| + 2\epsilon M$$

Now, Grönwall's inequality (Lemma 15) gives us the bound

$$\|\alpha(t)\| \leq 2\epsilon t M e^{Lt} \leq 2\epsilon t_{\max} M e^{L t_{\max}} = O(\epsilon) \tag{53}$$

Since $t_{\max}, L, M$ are fixed, we can choose $\epsilon_0$ such that $\epsilon_0 < \epsilon_1$ and $2\epsilon_0 t_{\max} M e^{L t_{\max}} < \epsilon_2$, which ensure that for all $x_0 \in \mathcal{C}, \epsilon < \epsilon_0$ and $t < \min(t_1, t_{\max})$, the point $x_1^{(\epsilon)}(t)$ is in the interior of $\mathcal{K}_2$. Therefore, if $t_1 \leq t_{\max}$ then $\lim_{t \to t_1} \|x_1^{(\epsilon)}(t)\| \in \mathcal{K}_2$, which contradicts the extensibility result. Thus, $t_1 > t_{\max}$, and hence flow maps for (52) exists for all $0 \leq \epsilon \leq \epsilon_0$ and $0 \leq t \leq t_{\max}$.

Now, we end with the remark that since $F_1 = F$ and $G_1 = G$ in $\mathcal{K}_2$, the flow map of (52) is a flow map for (51) inside $\mathcal{K}_2$, and therefore, solutions to (51) exist for all $x_0 \in \mathcal{C}, 0 \leq \epsilon \leq \epsilon_0$ and $0 \leq t \leq t_{\max}$.

Lastly, we will comment on value of $M$. Let $G$ be $L_1$-Lipschitz on $\mathcal{K}_2$, and let

$$M' = \max_{0 \leq t \leq t_{\max}, x_0 \in \mathcal{C}} \|G(x^{(0)}(t), t)\|$$

Then $M \leq M' + \epsilon_0 L_1$. Therefore, we can just choose $\epsilon_0$ small enough so that $M \leq 2M' + 1$, which enforces the constants in $O(\cdot)$ notation to depend only on $L, M'$ and $t_{\max}$.

$\square$

**Lemma 14.** *Consider the ODE's*

$$\frac{d}{dt}x(t) = F(x(t), t) + \epsilon G(x(t), t) \tag{54}$$

$$\frac{d}{dt}y_0(t) = F(y_0(t), t)$$

$$\frac{d}{dt}y(t) = F(y(t), t) + \epsilon G(y_0(t), t)$$

*such $F, G : \mathbb{R}^n \times \mathbb{R} \to \mathbb{R}^n$ are in $C^{r+1}$. Let $\mathcal{C} \subseteq \mathbb{R}^n$ be a compact set, and suppose that solutions to (54) exist for all $x_0 \in \mathcal{C}$. Let $T^x(x_0), T^{y_0}(x_0)$, and $T^y(x_0)$ be the time $t_{\max}$-flow map corresponding to this ODE for initial values $x(t) = y_0(t) = y(t) = x_0$.*

*Then as $\epsilon \to 0$, the maps $T_t^x$ and $T_t^y$ are $O(\epsilon^2)$ uniformly close over $\mathcal{C}$ in $C^r$ topology, for all $t \in [0, t_{\max}]$. The constants in the $O(\cdot)$ depend on $\max_{0 \leq k \leq r+1, x_0 \in \mathcal{C}, 0 \leq t \leq t_{\max}} \|D^k F(x, t)|_{x = y_0(t)}\|$ (the first $r+1$ derivatives of $F$ on the $y_0$-trajectories) and $\max_{0 \leq k \leq r, x_0 \in \mathcal{C}, 0 \leq t \leq t_{\max}} \|D^k G(x, t)|_{x = y_0(t)}\|$, (the first $r$ derivatives of $G$ on the $y_0$-trajectories).*

*Proof.* Let $F_\epsilon(x, t) = F(x, t) + \epsilon G(x, t)$, and let $T_t^\epsilon(x_0)$ denote the flow map of (54) starting at $x_0$. From (43), there is a polynomial $P = P_{i_1, \ldots, i_r}$ such that

$$\frac{d}{dt}\partial_{i_1 \cdots i_r} T_t^x(x_0) = \sum_{i=1}^d \partial_i F_\epsilon(x(t), t) \partial_{i_1 \cdots i_r} T_{t,i}^\epsilon + P(DF_\epsilon, \ldots, D^r F_\epsilon, DT_t^x, \ldots, D^{r-1} T_t^x) \tag{55}$$

On the other hand, applying (43) to $y_0$ gives

$$\frac{d}{dt}\partial_{i_1 \cdots i_r} T_t^{y_0}(x_0) = \sum_{i=1}^d \partial_i F(y_0(t), t) \partial_{i_1 \cdots i_r} T_{t,i}^{y_0} + P(DF, \ldots, D^r F, DT_t^{y_0}, \ldots, D^{r-1} T_t^{y_0})$$

We will now show that these two trajectories are $O(\epsilon)$ uniformly close by induction on $r$. Note that the base case ($r = 0$) is proved in Lemma 13. We will first show that

$$\|P(DF_\epsilon, \ldots, D^r F_\epsilon, DT_t^x, \ldots, D^{r-1} T_t^x) - P(DF, \ldots, D^r F, DT_t^{y_0}, \ldots, D^{r-1} T_t^{y_0})\| = O(\epsilon)$$

Since $P$ is a fixed polynomial that depends on $i_1, \ldots, i_r$, to show the above, we only need to show that the coordinates are $O(\epsilon)$ close, for small enough $\epsilon$.

$$\|D^k F_\epsilon(x(t), t) - D^k F(y_0(t), t)\| \leq \|D^k F_\epsilon(x(t), t) - D^k F(x(t), t)\| + \|D^k F(x(t), t) - D^k F(y_0(t), t)\|$$
$$\leq \epsilon \|D^k G(x(t), t)\| + \|x(t) - y_0(t)\|(2N_{k+1} + 1)$$
$$\leq O(\epsilon(2M_k + 2N_{k+1} + 2))$$

where $N_{k+1} = \sup_{x_0 \in \mathcal{C}, 0 \leq t \leq t_{\max}} \|D^{k+1} F(x, t)|_{x = y_0(t)}\|$ and $M_k = \sup_{x_0 \in \mathcal{C}, 0 \leq t \leq t_{\max}} \|D^k G(x, t)|_{x = y_0(t)}\|$. The second inequality follows since the base case

(Lemma 13) implies that $\|x(t) - y_0(t)\| = O(\epsilon)$, and since $D^{k+1}F$ is continuous, it follows that for small enough $\epsilon$, $\|D^{k+1}F|_{(x,t)}\| \leq 2N_{k+1} + 1$, for all $x$ such that $\|x - y_0(t)\| = O(\epsilon)$. Similarly, note that for small enough $\epsilon$, $\|D^k G(x(t), t)\| \leq 2M_k + 1$, since $G$ is $C^k$. Therefore, $\|D^k F_\epsilon(x(t), t) - D^k F(y_0(t), t)\| = O(\epsilon)$, where constants in $O(\cdot)$ depend $M_k$ and $N_{k+1}$.

To simplify notation, let $\alpha(t) = \frac{d}{dt}\partial_{i_1 \cdots i_r}(T_t^x - T_t^{y_0})$. Then,

$$\frac{d}{dt}\alpha(t) = \frac{d}{dt}\partial_{i_1 \cdots i_r}(T_t^x - T_t^{y_0})$$

$$= \sum_{i=1}^d \partial_i F_\epsilon(x(t), t)\partial_{i_1 \cdots i_r} T_{t,i}^x - \sum_{i=1}^d \partial_i F(y_0(t), t)\partial_{i_1 \cdots i_r} T_{t,i}^{y_0} + O(\epsilon)$$

$$= \sum_{i=1}^d \partial_i F_\epsilon(x(t), t)\partial_{i_1 \cdots i_r}(T_{t,i}^x - T_{t,i}^{y_0}) + \sum_{i=1}^d (\partial_i F_\epsilon(x(t), t) - \partial_i F(y_0(t), t))\partial_{i_1 \cdots i_r} T_{t,i}^{y_0} + O(\epsilon)$$

$$= DF_\epsilon(x(t), t)\partial_{i_1 \cdots i_r}(T_t^x - T_t^{y_0}) + (DF_\epsilon(x(t), t) - DF(y_0(t), t))\partial_{i_1 \cdots i_r} T_t^x + O(\epsilon)$$

$$= DF_\epsilon(x(t), t)\alpha(t) + (DF(x(t), t) - DF(y_0(t), t) + \epsilon G(x(t), t))\partial_{i_1 \cdots i_r} T_t^{y_0} + O(\epsilon)$$

$$\Rightarrow \frac{1}{2}\frac{d}{dt}\|\alpha\|^2 \leq \|DF_\epsilon(x(t), t)\|\|\alpha\|^2 + O(\epsilon(N_2 + M_0))\|\partial_{i_1 \cdots i_r} T_t^{y_0}\| + O(\epsilon)$$

$$\Rightarrow \frac{d}{dt}\|\alpha\| \leq \|DF(x(t), t)\|\|\alpha\| + O(\epsilon)$$

$$\leq (2N_1 + 1)\|\alpha\| + O(\epsilon)$$

Now, Grönwall's inequality (Lemma 15) gives us the bound,

$$\|\alpha(t)\| \leq t_{\max} e^{N_1 t_{\max}} O(\epsilon) = O(\epsilon)$$

The constants in the last $O(\cdot)$ notation depend on $t_{\max}$, $N_k$ for $0 \leq k \leq r + 1$ and $M_k$ for $0 \leq k \leq r$. This tells us that

$$\|T_t^x - T_t^{y_0}\|_{C^r} = O(\epsilon) \tag{56}$$

Now, note that $T_t^y$ satisfies

$$\frac{d}{dt}y(t) = F(y(t), t) + \epsilon G(y(t), t) + \epsilon(G(y_0(t), t) - G(y(t), t))$$

$$\implies \frac{d}{dt}y(t) = F(y(t), t) + \epsilon G(y(t), t) + \epsilon^2 H(y(t), t)$$

where $H(y, t) = \frac{1}{\epsilon}(G(y_0(t), t) - G(y(t), t))$. Consider the system of ODEs

$$\frac{d}{dt}y(t) = F_\epsilon(y(t), t) + \gamma H(y(t), t) \tag{57}$$

Note that when $\gamma = 0$, $T_t^x$ is the flow map for this system, and when $\gamma = \epsilon^2$, $T_t^y$ is the flow map for this system. Therefore, applying (56) for the system (57), we get

$$\|T_t^x - T_t^y\|_{C^r} = O(\gamma) = O(\epsilon^2)$$

where the constants in $O(\cdot)$ notation depend on $\sup_{0 \leq k \leq r, x_0 \in \mathcal{C}, 0 \leq t \leq t_{\max}} \|D^{k+1}F_\epsilon(x(t), t)\|$ which is bounded by $\max_{0 \leq k \leq r}(2N_{k+1} + 1)$ for small $\epsilon$, and $M_k' = \sup_{0 \leq k \leq r, x_0 \in \mathcal{C}, 0 \leq t \leq t_{\max}} \|D^{k+1}H(x(t), t)\|$. Using the definition of $H$,

$$\|D^k H(x(t), t)\| = \frac{1}{\epsilon}\|D^k G(y_0(t), t) - D^k G(x(t), t)\|$$

$$\leq \frac{1}{\epsilon}\|y_0(t) - x(t)\|(2M_{k+1} + 1)$$

$$= \frac{1}{\epsilon} \cdot O(\epsilon) \cdot (2M_{k+1} + 1) = O(1)$$

where the constant in the $O(\cdot)$ depends on $M_0, \ldots, M_{r+1}$ and $N_1, \ldots, N_{r+1}$. This proves the dependence in $O(\cdot)$ notation as stated in the statement, completing the proof. $\qquad \square$

**Corollary 1.** Consider the ODE
$$\dot{x} = Ax + \epsilon g(x, t)$$
such that $\|A\| = 1$ and $g$ has bounded $(r+1)^{th}$ derivatives on a compact set $\mathcal{C}$. Let $T^x$ be the flow map corresponding to this ODE. For fixed $x_0$, let $y_0, y_1$ be functions satisfying
$$\dot{y_0} = Ay_0$$
$$\dot{y_1} = Ay_1 + g(y_0(t), t)$$
such that $y_0(0) = x_0$ and $y_1(0) = 0$. Consider the flow map $T^y : \mathbb{R} \times \mathbb{R}^n$ such that $T^y(t, x_0) = y_0(t) + \epsilon y_1(t)$. Then, the maps $T_t^x$ and $T_t^y$ are $O(\epsilon^2)$ uniformly close over $\mathcal{C}$ in $C^r$ topology, for all $t \in [0, 2\pi]$. The constants in the $O(\cdot)$ depend on $\|A\|$ and the first $r$ derivatives of $g$ on the trajectories $x(t) = e^{At}x_0, x_0 \in \mathcal{C}$.

This follows directly from Lemma 14, after noting $\dot{y} = Ay_0 + \epsilon Ay_1 + \epsilon g(y_0(t), t) = Ay + \epsilon g(y_0(t), t)$. Note that $F(x) = Ax$ is a linear function, so derivatives of $F$ are bounded, and the $y_0$ trajectories can be computed easily.

### D.6   Grönwall lemma

The following lemma is very useful for bounding the growth of solutions, or errors from perturbations to ODE's.

**Lemma 15** (Grönwall). *If $x(t)$ is differentiable on $t \in [0, t_{\max}]$ and satisfies the differential inequality*
$$\frac{d}{dt}x(t) \le ax(t) + b,$$
*then*
$$x(t) \le (bt + x(0))e^{at}$$
*for all $t \in [0, t_{\max}]$.*