# OpenReview forum: "Universal Approximation Using Well-Conditioned Normalizing Flows"
_NeurIPS.cc/2021/Conference — NeurIPS 2021 Poster_

### Official Review · Reviewer_NQpX · 2021-07-12

**Rating:** 7
**Confidence:** 3

**Summary:**

Authors answer the question: "Can well-behaved distributions be approximated by an afﬁne coupling ﬂow with a well-conditioned Jacobian?" They answer by the affirmative, proving that log-concave distributions belong the the class of distributions that can be approximated by such flows, if the input distribution is padded with independent Gaussians.

**Limitations And Societal Impact:**

I wouldn't say that the limitations of this work are clearly discussed.

**Main Review:**


## Strengths
The main strength of this work is its theoretical soundness and the ability to overcome the limitations of previous work, namely the ill-conditioned Jacobian yielded by their assumptions.
The bound only depends on properties of the distribution (and does not worsen as epsilon gets small, as opposed to Koehler et al., 2020), which is satisfying.
Additionally, the independent Gaussian padding assumption that is made backs previous empirical results that have shown their usefulness in terms of conditioning of the flow.

## Weaknesses
Perhaps that are two minor limitations in this work that could potentially be alleviated.
1/ Relation with practical use of affine-coupling flows. This entire approach is based on augmenting the space with an auxiliary variable $v$ (via padding). It has apparently been shown that in practice it does indeed help with the conditioning of the flow (Koehler et al., 2020), yet I don't believe that it is a common practice. It would be nice to remind us what does better conditioning brings for practitioners. Is this work claiming that we should in practice pad the input?
2/ Discussion of other normalizing flows. I may be missing something, but it seems that From Equation 11 a similar result could be proved for continuous normalizing flows and similarly for Equation 12 and residual flows.
3/ Relationship with optimal transport map.
If I understand correctly, this work is claiming that afﬁne coupling ﬂows are universal approximator of the flow-map from underdamped Langevin dynamics. Why are underdamped Langevin dynamics a natural choice? As opposed to the optimal transport map for instance, which seems quite natural as Theorem 1 is stated in terms of an upper bound on the Wasserstein distance. Is it the same map with some assumptions? Is there any empirical evidence that in practice afﬁne coupling ﬂow approximate this Langevin flow-map?

## Clarity
The paper is rigorously written and quite well organized. I believe nonetheless that the flow could still improved to expand the audience of the work. In particular, Theorem 2 could need a bit more background or explanation. Also it is a bit unclear why and where we need Hénon maps. More generally, as Section 5 is a sketch of proof (the full proof being in the appendix), it would be worth appealing perhaps a bit more to the intuition of the reader at times (even though I acknowledge there are already some good effort on that regard).

## Relation to prior work
It would be perhaps relevant to mention theoretical work on the representational capacity of other class of normalizing flows (e.g. Huang et al. 2020).

## Additional feedback
89: Not clear how `underdamped Langevin dynamics' relates to log-concave distribution from this sentence.
142: "don't"
166: Shouldn't this be stated as a proper result?
168-169: This intuition is useful. The connection with the Gaussian padding (as a Gaussian prior for the velocity) could be stressed.
183: It would be useful to briefly explain why L is an upper bound of the KL divergence.
282: Is there a practical way to compute the value of $\tau$? That can be useful as it informs us on the number of layers required.
297: Why are these functions polynomial in $\cos(\Omega t)$ and $\sin(\Omega t)$?
306: Would be quite useful to write explicitly the associated non-linearities $s,t$ for both layers. Is appears that $s$ is always of the form $s=(I_d + \tilde{s})$ due to the Euler discretization.


**Time Spent Reviewing:**

4

---

> ### Author Response · Authors · 2021-08-10
> **Thank you for the thoughtful review!**
>
> We thank the reviewer for the feedback and kind words! We are very happy you appreciated our contributions and enjoyed reading our paper!
>
> We respond to some of your concrete questions.
>
> **Re: why is padding not more common practice**
>
> We’re not sure why padding is not more frequently used! [Koehler et al. 2020] indeed observed that it frequently helps empirically (both on real and synthetic data), and our paper provides a formal proof that this indeed is the case for the log-concave case.
>
> **Re: other flows like continuous normalizing flows and residual flows:**
>
> You are right, thank you for pointing out our results also apply to other families of models! See our common response for a sketch of this observation.
>
> **Re: “Relationship with optimal transport map. If I understand correctly, this work is claiming that afﬁne coupling ﬂows are universal approximator of the flow-map from underdamped Langevin dynamics. Why are underdamped Langevin dynamics a natural choice?”**
>
> The optimal transport map is different. As far as we can tell, there is no simple way to express it as an affine coupling map. In contrast, the form of the underdamped Langevin equation lends itself to approximation using an affine coupling map (through Henon-like maps), because of its resemblance to a Hamiltonian system of ODE’s.
>
> **Re: “Is there any empirical evidence that in practice afﬁne coupling ﬂow approximate this Langevin flow-map?”**
>
> It is a very interesting question for future empirical work to check whether affine coupling flows approximate some Langevin-like process in practice !
>
> **Re: “In particular, Theorem 2 could need a bit more background or explanation.**
>
> We agree! Unfortunately, we had to leave some details out due to space restrictions. Space permitting, we will add some more details.
>
> **Re: “Also it is a bit unclear why and where we need Hénon maps”**
>
> The final affine coupling model that we construct arises from discretizing Eq (11) in Lemma 3. Essentially, if we look at the discretized updates for $x_n$ and $v_n$ (details in Appendix D.2), these have the form of Henon-like maps (as defined in Definition 7), and these can be written as affine coupling layers.
>
> **Re: other flows like continuous normalizing flows and residual flows:**
>
> Thanks for the suggestion! Please see our common response for some comments on this.
>
> ------------
> **Re: “Additional Feedback”**
>
> Thanks for taking your time to point these out carefully!
>
> **89:** *Not clear how `underdamped Langevin dynamics' relates to log-concave distribution from this sentence.*
>
> The relation is that the underdamped langevin dynamics has fast convergence for log-concave distributions (see Ma et. al. for details)
>
> **166:** *Shouldn't this be stated as a proper result?*
>
> The relevant result associated with this definition is stated in theorem 2.
>
> **183:** *It would be useful to briefly explain why L is an upper bound of the KL divergence.*
>
> Note that the expression for L in Theorem 2 consists of the KL divergence plus a positive term (because S is positive definite), which is why it is an upper bound on the KL divergence.
>
> **282:** *Is there a practical way to compute the value of $\tau$? That can be useful as it informs us on the number of layers required.*
>
> The approximation error in Lemma 5 is $O(\tau)$, and the exact value of $\tau$ to be chosen depends on the constants inside the $O(\cdot)$ (refer to Appendix C). A rough estimate would be $\tau = 1/\epsilon^{2\kappa}$. As for the number of layers required, a rough bound on the depth of the constructed affine-coupling flow model is $\text{polylog}(\epsilon)/ \epsilon^{O(\kappa)}$. Thanks for these questions, we will include the estimates in the camera ready version !
>
> **297:** *Why are these functions polynomial in cos⁡(Ωt) and sin⁡(Ωt)?*
>
> The reason is that polynomials in sin/cos are a rich enough class to approximate the desired ODE. Please refer to Appendix B for details.

---

> > ### Comment · Reviewer_NQpX · 2021-08-18
> > **Reply to rebuttal**
> >
> > I Thank the reviewers for the clarifying comments. I am glad that the authors confirmed that their proof can straightforwardly be applied to  show similar results for continuous and residual flows. I would still be very curious to know whether the learnt affine coupling flow actually approximates the Langevin flow-map. If so it would strengthen this work. Additionally, the log-concavity assumption does restrict quite strongly the class of distributions being encompassed in the theorem, and the paper does not consider (even empirically) any larger class of distributions.
> >
> > I believe that overall this submission is good and relevant but because of the aforementioned limitations I am keeping my rating at 7.

---

### Official Review · Reviewer_Jjda · 2021-07-16

**Rating:** 7
**Confidence:** 3

**Summary:**

The paper gives a proof of universal approximation by well-conditioned (affine-coupling) normalizing flows of arbitrary log-concave distribution. The main difference from previous results on universal approximation by such flows is that transformation is well-conditioned. The proof is constructive and potentially allows to extract the number of flow layers required to achieve given accuracy.

**Limitations And Societal Impact:**

yes

**Main Review:**

The paper considers the question of  universal approximation property of affine-coupling normalizing flows. Previous results on this subject established universal approximation for ill-conditioned flows that have their condition number diverge with approximation quality. The authors attack the problem by restricting the target distribution to be log-concave with bounded log-probability Hessian. Using SDE convergence results they define an ODE (equivalent to Langevine SDE), that provides approximate distribution via a well-conditioned transform. Further, they extend a known equivalence (Henon map <-> Simplectic flow) to recast previous approximation into the form of a generalized Henon map. The discrete version of this map has the form of a sequence of (pairs of) affine-coupling flows. This allows the authors to establish the universal approximation property of affine-coupling normalizing flows.


The contribution is original and very relevant for practical applications. The paper gives theoretical grounds for using padding of data distribution with Gaussians for the purpose of approximating it with well-conditioned NF.
The paper is well written, giving most definitions and qualitative proofs in the main text, which makes it easy to follow. The authors did a really good job in presenting the complex proofs in a very limited space.


Remarks:
1. in line 174 the equation U(x) >=	\lambda I is a bit confusing, because U is scalar and I is a unit matrix
2.  Even though it may be not physically possible given space constraints, but it would be helpful to add a diagram, representing different pieces that were used in the proof, with e.g. arrows showing connections/equivalences.
3. I found it hard (not being as expert in the material) to track the connection between "condition parameters" k (line 107), lambda (line 174) and rho (eq (2)): may be adding remarks on how they connect to each other around the places where they are used could help with that.
4. Finally, even though the number of flow layers required to achieve given accuracy can be extracted from the proof, having an explicit formula would be helpful for practitioners.


****************************************************
Post-rebuttal

I'd like to thank the authors for their response and will keep my already high score the same.


**Time Spent Reviewing:**

4

---

> ### Author Response · Authors · 2021-08-10
> **Thank you for the thoughtful review!**
>
> We thank the reviewer for the feedback and kind words! We are very happy you appreciated our contributions and enjoyed reading our paper!
>
> Responding to your remarks:
>
> **Re: “in line 174 the equation U(x) >= \lambda I is a bit confusing, because U is scalar and I is a unit matrix”**
>
> This is a typo; it should read $nabla^2 U \ge \lambda I$.
>
> **Re: the connection between "condition parameters" k (line 107), lambda (line 174) and rho (eq (2))**
>
> Thank you for this suggestion! Due to space limitations, we tried to be terse in explaining our proof steps as possible, but even so, we will try to improve the writing so as to be more expository about the relevance of these parameters in the revised version.
>
> **Re:  number of flow layers**
>
> A simple bound on the depth of the constructed affine-coupling flow model is $\text{polylog}(\epsilon)/ \epsilon^{O(\kappa)}$. We are happy to include the computation in the camera ready version.

---

### Official Review · Reviewer_PyGo · 2021-07-16

**Rating:** 7
**Confidence:** 3

**Summary:**

This paper proves that _well-conditioned_ affine-coupling-based normalizing flows can approximate any _log-concave_ distribution. The proof techinques used in this paper combine underdamped Langevin dynamics and methods from dynamical systems in a novel way. The actual result in this paper involves approximating, to arbitrary precision, a joint distribution comprised of the original target padded with Gaussian noise of the same dimension. Then, universality is implied by the fact that the marginal along the dimensions of the original target distribution must be equal.

**Limitations And Societal Impact:**

I think there could have been more discussion about the practical implications of the results beyond just stating that likelihood-based training requires stability in the flow. How can we use the insights here to train flows better? Can we make some recommendations for generic distributions, or can we only say that normaliizing flows can stably approximate log-concave distributions?

I have no concerns about the societal impact.

**Main Review:**

Overall, I found this paper to be an interesting read and would recommend it for inclusion in the conference. I'll provide further details along the standard review axes below.

### Originality

Studying the approximation properties of $d$-dimensional normalizing flows by embedding in a $2d$-dimensional space is not entirely novel, but the result achieved and the techniques used to do so appear original as far as I am aware. I will note a couple omissions in the related work, however:
1. The term "normalizing flow" was actually coined in _Density estimation by dual ascent of the log-likelihood_  [Tabak & Vanden-Eijnden, 2010], with normalizing flows being further developed in the follow-up _A family of nonparametric density estimation algorithms_ [Tabak & Turner, 2013]. Although Rezende & Mohamed, 2015 and Dinh et al., 2014 may have popularized the notion of flows in the machine learning community, it would be remiss to say that the inception of the ideas were in these papers alone.
2. Two recent works, _Relaxing bijectivity constraints with continuously-indexed normalizing flows_ [Cornish et al., 2020] and _Understanding and mitigating exploding inverses in invertible neural networks_ [Behrmann et al., 2021] study the theoretical and empirical pitfalls, respectively, of using ill-conditioned normalizing flows. The former in particular notes that the (Bi-)Lipschitz constant of a normalizing flow -- and therefore its condition number -- must necessarily approach infinity if the flow distribution is to approximate a distribution with non-trivial topology to arbitrary precision. The latter provides a complementary empirical view of the matter, demonstrating that normalizing flows quite often display numerical non-invertibility when fully trained. Including these results in the related work may help bolster your motivation even further.

### Quality

Although I did not check this closely, the mathematical results in the paper appear to be correct. This paper does a good job of bringing together ideas from different fields to produce a proof. My one criticism here though is that the result is not looked at critically - it is only mentioned that the result is limited to log-concave distributions, without further discussions of the limitations of the paper overall which I've included in the Significance section of the review.

### Clarity

I found the paper to be clearly written, and I very much appreciated the intuitive proof sketches provided throughout.

### Significance

I think that demonstrating how to show the main result is the most important part of the paper, as opposed to the result itself. What I mean is that the combination of techniques provided in this work may allow others to prove stronger results about the approximation properties of certain classes of flows. I do not find the result itself to be particularly shocking, as log-concave distributions do not appear to be a particularly difficult class of distributions to approximate. Furthermore, this work does not really consider the practical implications of using normalizing flows for generic distributions, only noting that they can approximate log-concave distributions without blowing up. I would have liked to see some study or discussion on the practical implications of this result, as for now I cannot really think of any besides this paper possibly inspiring future works with its mathematical techniques.

### Miscellaneous Comments

- How restrictive is the condition $I_d \leq - \nabla^2 \log p(x) \leq \kappa I_d$? What happens if $-\nabla^2 \log p(x)$ has eigenvalues between $0$ and $1$, and how often might we expect this to occur?
- I find it quite counterintuitive that $\mathcal C$ can be made arbitrarily large by making $\epsilon$ arbitrarily small - can you comment on this further?
- Is it required, strictly speaking, to pass the proof to a $2d$-dimensional space instead of the original $d$-dimensional space, and why? Is there some smaller dimension where things might still work properly?

**Time Spent Reviewing:**

4

---

> ### Author Response · Authors · 2021-08-10
> **Thank you for the thoughtful review!**
>
> We thank the reviewer for the feedback and kind words! We are happy you appreciated our paper!
>
> **Regarding the two related works:** Thanks for providing the references and suggestions! We will add a discussion in the camera-ready version.
>
> **Regarding the practical implications of the work:** We focused less on that in this paper, as the prior work [Koehler et al. 2020] already provided empirical evidence for the usefulness of Gaussian padding during training. We share your feeling that the mathematical ideas in this paper are of separate interest from the main result itself.
>
> Finally, we include some comments on removing the log-concavity assumption in the common response above.
>
> -------
>
> Answers to the miscellaneous comments follow:
>
> **Re: “How restrictive is the condition $I_d\le -\nabla^2 \log p(x)\le \kappa I_d$?”**
>
> We comment on the restrictiveness of log-concavity in the common response. Further, note that we can deal with any distribution with bounded condition number in the usual sense ($a I_d\le -\nabla^2 \log p(x)\le \kappa a I_d$) by simple rescaling (or equivalently, running underdamped Langevin with stationary distribution $e^{-a||x||^2/2}$ instead).
>
> **Re: “I find it quite counterintuitive that C can be made arbitrarily large by making ϵ arbitrarily small - can you comment on this further?”**
>
> $\epsilon$ denotes the amount of mass we are not capturing; letting it go to 0, we need the set to become larger to capture more of the distribution, eventually covering the entire space when $\epsilon = 0$
>
> **Re: “Is it required, strictly speaking, to pass the proof to a 2d-dimensional space instead of the original d-dimensional space, and why? Is there some smaller dimension where things might still work properly?”**
>
> 2d dimensions is required for our approach, as underdamped Langevin requires augmenting with a velocity variable which has the same dimension as the position.
>
> **Re: “How can we use the insights here to train flows better?”**
> Recapping from above, the benefits of Gaussian padding have already been noted in [Koehler et al. 2020] on real-life data and synthetic data, which is certainly not log-concave.

---

> > ### Comment · Reviewer_PyGo · 2021-08-13
> > **Rebuttal Response**
> >
> > Thanks for your clarifications, and again thanks for the interesting paper. I will keep my score at the same level, which was already a 7.

---

### Official Review · Reviewer_Z2tC · 2021-07-21

**Rating:** 7
**Confidence:** 2

**Summary:**

In this paper, authors provide the first proof that well conditioned affine coupling flows can approximate any log concave distribution.The authors employ stochastic differential equations and structured dynamical systems to construct their proof, uncovering some unknown connections.

**Limitations And Societal Impact:**

I don't think this applies.

**Main Review:**

Coupling flows have been one of the most widespread used members of the flow family; due to the easy of implementation and computation, they have attracted more attention than other variants. However, as mentioned elsewhere and pointed out by authors, the theoretical understanding of their representation capabilities have been limited. This paper provide the first satisfactory result to that front: well conditioned affine coupling flows can represent any log-concave density.

Overall, the paper is very well written. I applaud the authors for their efforts to make a paper that is basically a long proof to be so readable. Clearly, the paper has been tailored to be as accessible to computer scientists as possible; I commend the authors for their efforts.

I believe the paper has sufficient merit; a valid justification for the use of affine coupling flows is a welcome addition to the literature. Moreover, the proof technique and potentially help future analysis.


**Time Spent Reviewing:**

4-5 hours

---

> ### Author Response · Authors · 2021-08-10
> **Thank you for the thoughtful review!**
>
> We thank the reviewer for the feedback and kind words! We are happy you appreciated our efforts to make the paper accessible!

---

### Official Review · Reviewer_5zdr · 2021-07-23

**Rating:** 6
**Confidence:** 3

**Summary:**

The authors study universal approximation with well-conditioned affine coupling networks. They proved that the log-concave distributions can be approximated by an affine coupling flow with a well-conditioned Jacobian. To prove this argument, they study the connections between stochastic differential equations, dynamical systems and affine coupling flows. In their construction, Gaussian padding was adopted, as suggested by the empirical observation from [Koehler et al., 2020].

**Limitations And Societal Impact:**

Yes, the authors have adequately addressed the limitations and potential negative societal impact of their work.

**Main Review:**

Originality: The related works are adequately cited. The novelty of this paper is high. The universal approximation results in this paper, as mentioned in the above summary part, will certainly help us have a better understating of affine coupling networks from a theoretical way. I have checked the technique parts and find that the proofs are solid. I think this is a significant contribution to machine learning immunity. My only concern is that, the authors only consider the universal approximation for log-concave distributions, which make the contribution of this paper limited. It will be interesting to explore the universal approximation of affine coupling networks for more general distributions in the future.

Quality: This paper is technically sound.

Clarity: This paper is clearly written and well organized. I find it easy to follow.

Significance: I think the results in this paper are significant, as explained above.

**Time Spent Reviewing:**

5 hours

---

> ### Author Response · Authors · 2021-08-10
> **Thank you for the thoughtful review!**
>
> We thank the reviewer for the feedback and kind words! We are happy you appreciated the novelty and contributions in our paper! Our response to your concern about the restrictiveness of log-concavity is in the common response above.

---

### Official Review · Reviewer_yz9H · 2021-07-24

**Rating:** 7
**Confidence:** 3

**Summary:**

This paper proves a universal approximation theorem of well-conditioned affine coupling flows for log-concave probability distributions. This extends previous approximation results with ill-conditioned flows. The paper takes a different proof approach with previous works: it constructs an underdampled Langevin dynamics, which is a well-conditioned pushforward between the Gaussian-augmented data distribution and standard Gaussian. Then, the paper shows that affine coupling blocks can simulate the Langevin dynamic.

**Limitations And Societal Impact:**

This work does not have foreseeable negative societal impact.

**Main Review:**

The most significant contribution of this paper is it brings in the powerful SDE tool to study affine coupling flows. The proof technique is novel. It's the first time I have seen such an approach. I think the method can be used for analyzing other problems, e.g., extended beyond log-concave distributions. I didn't check the proof in detail, but the idea seems valid. The quality of the work is good. The theorems and background knowledge are nicely organized. The presentation is clear.

In terms of significance, one limitation of the proposed approach is it seems to be limited to affine coupling flows, since the construction of underdamped Langevin dynamics is specially designed to meet the two-part structure of affine coupling flows. It would be more interesting if the authors can demonstrate the idea for other types of flows, e.g., residual flows.

Another result I would like to see is the practical implications of the theoretical result. Currently it is not discussed in the paper. For example, can the theoretical result guides us to improve the conditioning of affine coupling flows, or can the Langevin dynamics be used to construct more efficient flows?

The idea of ODEs and SDEs are also extensively used in Neural ODE, FFJord, residual flow, etc.. It is also interesting to see the connection between the proposed method and the neural ode family of methods. Can we compare their representation capacity / efficiency ?

Post rebuttal: I am satisfied with the authors' feedback regarding to the necessity of log-concavity and applicability to other types of flows. I'm increasing my score to 7.

**Time Spent Reviewing:**

3

---

> ### Author Response · Authors · 2021-08-10
> **Thank you for the thoughtful review!**
>
> We thank the reviewer for the feedback and kind words! We are happy you appreciated the connections to SDEs and the proof techniques! We answer your questions now:
>
> **Re: other flows like continuous normalizing flows and residual flows:**
>
> You are right, thank you for pointing out our results also apply to other families of models! See our common response for a sketch of this observation.

---

### Official Review · Reviewer_rkxh · 2021-07-24

**Rating:** 7
**Confidence:** 4

**Summary:**

The paper proves that log-concave distributions can be approximated by normalizing flows defined by affine-coupling models and starting from Gaussian distributions. The special feature of the result, distinguishing it from earlier universal approximation results for this type of flows, is that the constructed map is well-conditioned. The proof first approximates the target distribution using a Langevin dynamics, and then shows that this dynamics can be approximated by affine-coupling models.

**Limitations And Societal Impact:**

-

**Main Review:**

This is an interesting and technically complex paper that reveals some interesting connections between normalizing flows, Langevin evolution, and earlier research on Hénon maps as universal approximators for symplectic diffeomorphisms. While there are already several papers on universal approximation with normalizing flows, this paper poses and resolves the question of finding a well-conditioned flow (albeit under relatively restrictive conditions on the target distribution - log-concavity and Gaussian padding). The proposed (rather complex, two-stage) construction combines ideas of the Langevin evolution and earlier research on approximating Hamiltonian flows. The paper is clearly written and includes a sketch of the proof that explains its main points rather well. I think that the paper brings some novel ideas to the study of approximation properties of normalizing flows.

If I understand correctly, the well-conditioning in the paper refers only to the full flow map, so that the elementary affine-coupling layers comprising the flow can potentially be ill-conditioned (or ill-conditioning can accumulate because of the large number of these layers)?

I didn't quite understand at which point the Gaussian padding is used in the proof. Is it used in the Langevin convergence result (Lemma 1), because the target velocity distribution needs to be Gaussian?

Probable typos: the minus sign in the first equation in system (1), missing $\gamma$ in the covariance of $\xi_t$ in line 250.

Post-rebuttal: I thank the authors for answering my questions.





**Time Spent Reviewing:**

6

---

> ### Author Response · Authors · 2021-08-10
> **Thank you for the thoughtful review!**
>
> We thank the reviewer for the feedback and kind words! We are happy you found the connections uncovered in the paper interesting! We answer your questions now:
>
> **Re: “elementary affine-coupling layers comprising the flow can potentially be ill-conditioned”**
>
> For any $s < t$ in [0, T], we do have that the condition number of the flow map from time $s$ to time $t$ is well-conditioned, as given by Lemma 1. However, when we discretize with step size $\eta$ and approximate the flow map by affine-coupling layers, we only have a partial guarantee that every composition of layers representing complete steps of size $\eta$ is well-conditioned. We don’t explicitly have conditioning guarantees for individual layers, although we believe that they hold. Getting the conditioning bound would involve analysing the coefficients of polynomials F, G, J given by the ODE system in lemma 3 in detail.
>
> **Re: “didn't quite understand at which point the Gaussian padding is used in the proof. Is it used in the Langevin convergence result (Lemma 1), because the target velocity distribution needs to be Gaussian?”**
>
> Yes. The stationary distribution is jointly Gaussian in the position and velocity variables - the velocity variables being the padded variables. Hence we need the padding to be Gaussian to match.
>
> **Re: “Typos: the minus sign in the first equation in system (1), missing γ in the covariance of ξt in line 250.”**
>  You are right!

---

### Author Response · Authors · 2021-08-10
**Comments for All Reviewers**

We thank the reviewers for their detailed reading of the paper and their various insightful comments!

We are encouraged at the overwhelming support for the paper, noting the originality of the proof techniques, the new connections between SDEs, affine couplings and Henon maps it uncovers, and the accessibility of the writing despite being technically very involved.

As a comment to all reviewers, we remark on a recurring question: **where is log-concavity crucially needed in the proofs, and could we generalize the results beyond log-concave distributions?** Log-concavity of the target distribution is necessary to ensure that the Jacobian of the transport map from latent to target distribution is well-conditioned. The overall proof strategy of leveraging a connection between symplectic diffeomorphisms and Langevin dynamics seems more general, and it would be interesting to see if our techniques can be extended beyond the log-concave setting.

To be more technically precise, by eq (21), bounding the condition number reduces to bounding $\nabla^2 (\ln p_t - \ln p)$, where $p_t$ is the probability distribution after flowing underdamped Langevin started at $p_0$ (the distribution we wish to approximate) with stationary distribution $p=\mathcal{N}(0,I)$ (standard Gaussian). For this, we require a handle on the Hessian of the log-pdf under convolution (Lemma 7), which we only know to hold for log-concave distributions. We could potentially extend beyond log-concavity, e.g. if we knew that $p_t$ was log-concave after some $t_0$, possibly incurring some exponential dependence in $t_0$.

Finally, we’d like to thank Reviewers yz9H and NQpX for essentially noticing that our proof implies that along the way, we prove well-conditioned universal approximation for other related normalizing flow models. Namely, our Lemma 1 proves that the flow map of underdamped Langevin dynamics is well-conditioned for all $t \in [0,T]$. However, it’s clear by [1] that underdamped Langevin dynamics is a continuous normalizing flow --- so the claim follows. Similarly, the particular affine coupling layers we construct (eq (12)) also form a residual block, so the claim also follows for residual flows [2].

[1] Ricky T. Q. Chen, Yulia Rubanova, Jesse Bettencourt, and David Duvenaud. Neural ordinary differential equations. Advances in Neural Information Processing Systems, 2018.

[2] Behrmann, J., Grathwohl, W., Chen, R. T., Duvenaud, D., & Jacobsen, J. H. Invertible residual networks. International Conference on Machine Learning, 2019.

---

### Decision · Program_Chairs · 2021-09-28

**Decision:**

Accept (Poster)

**Comment:**

The paper demonstrates that log-concave distributions can be well approximated by well-conditioned affine-coupling flows with Gaussian base density.

This paper is clearly written and technically sound. It provides novel insights and interesting connections between normalising flows, Langevin diffusions, Hénon maps and their (universal) approximation properties.

**Consistency Experiment:**

NeurIPS has a long history of experimentation. In 2014, NeurIPS ran an experiment in which 10% of submissions were reviewed by two independent committees to quantify the randomness in the review process. This year, we repeated a variant of this experiment to see how the quality of the review process has changed over time.  This paper was part of the experiment and was therefore assigned to two committees (consisting of reviewers, an Area Chair, and a Senior Area Chair) that reached independent decisions.  If both committees made the same recommendation, this recommendation was followed. If a single committee recommended acceptance, the paper was accepted (with the exception of a few cases in which the other committee identified what we considered a fatal flaw, e.g., an error in a key result).

Both committees reached the same decision: **Accept (Poster)**

The other committee assigned to the paper recommended **Accept (Poster)**.  You can find the other set of reviews, along with any follow up discussion with the authors here:
https://openreview.net/forum?id=qLpJ0VWRuWk